# SEEING IS NOT REASONING: MVPBENCH FOR GRAPH-BASED EVALUATION OF MULTI-PATH VISUAL PHYSICAL COT

## ABSTRACT

Understanding the physical world—governed by laws of motion, spatial relations, and causality—poses a fundamental challenge for multimodal large language models (MLLMs). While recent advances such as OpenAI o3 and GPT-4o demonstrate impressive perceptual and reasoning capabilities, our investigation reveals these models struggle profoundly with visual physical reasoning, failing to grasp basic physical laws, spatial interactions, and causal effects in complex scenes. More importantly, they often fail to follow coherent reasoning chains grounded in visual evidence, especially when multiple steps are needed to arrive at the correct answer. To rigorously evaluate this capability, we introduce **MVPBench**, a curated benchmark designed to rigorously evaluate visual physical reasoning through the lens of visual chain-of-thought (CoT). Each example features interleaved multi-image inputs and demands not only the correct final answer but also a coherent, step-by-step reasoning path grounded in evolving visual cues. This setup mirrors how humans reason through real-world physical processes over time. To ensure fine-grained evaluation, we introduce a **graph-based CoT consistency metric** that verifies whether the reasoning path of model adheres to valid physical logic. Additionally, we minimize shortcut exploitation from text priors, encouraging models to rely on visual understanding. Experimental results reveal a concerning trend: even cutting-edge MLLMs exhibit poor visual reasoning accuracy and weak image-text alignment in physical domains. Surprisingly, **RL-based post-training alignment—commonly believed to improve visual reasoning performance—often harms spatial reasoning**, suggesting a need to rethink current fine-tuning practices.

## 1 INTRODUCTION

Human comprehension of the world is fundamentally grounded in physical laws: objects fall when released, and liquids take the shape of their containers Spelke & Breinlinger (1992); Baillargeon (2004). Such physical regularities form the basis of our causal understanding Gopnik et al. (2004); Lake et al. (2017), and further link the chain of reasoning when solving complex problems. Recent advances appear to grasp this physical world that humans experience—a blitz of multimodal large language models (MLLMs) like OpenAI o3 OpenAI (2025), GPT4o OpenAI (2024), Gemini Deepmind (2024), InternVL3 Zhu et al. (2025), Kimi1.5 KimiTeam (2025) and many othersLiang et al. (2025); Zheng et al. (2025b) -all claiming *human-level physical reasoning* after a final reinforcement-learning (RL) post-training. Recent works Shao et al. (2024); Guo et al. (2025); Li et al. (2025b); Daxberger et al. (2025); Huang et al. (2025); Fan et al. (2024) show models describing panoramic scenes, solving game reasoning, even generating Chain-of-Thought (CoT) explanations. At first glance, it feels as thought **plug-and-play embodied intelligence is already on our doorstep.**

Full of eager expectation, we asked the lastest MLLMs a child-level physics question. *What is the direction of movement for the car?* Fig. 1(left) shows the setup. Surprisingly, GPT-4o responded with an incorrect prediction. Pushing further, we queried the thought chain of models. The failure patten was consistent: models *saw* the pixels but did not *reason* about forces, geometry, or causality.

"The Second Half," reminds us AI is entering a phase where evaluation outweighs trainingYao (2025). Yet current benchmarks used to "prove" spatial reasoning are a weak compass. Most

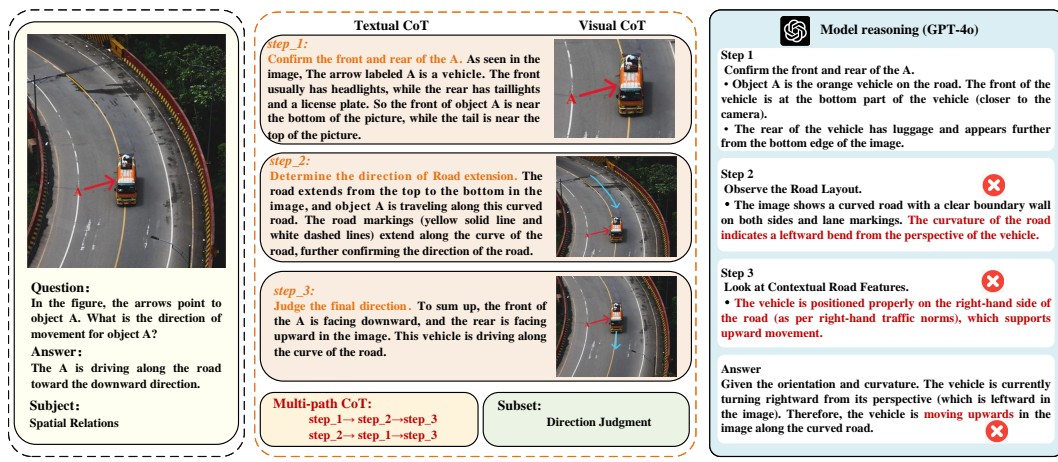

**Figure 1 A one-minute sanity check shatters the illusion *of spatial reasoning in MLLMs*.** Red arrows indicate objects and multiple reasoning chains are provided to capture diverse yet valid solution strategies.

rely on game-engine videos or CAD renderings whose textures and lighting barely resemble the messy real worldKang et al. (2024); Zheng et al. (2024). In addition, many questions are phrased so that a language-only model can guess the answer from commonsense priors, bypassing vision altogetherZhou et al. (2025); Yue et al. (2024); il Lee et al. (2025). Furthermore— almost none pair each intermediate visual cue with an explicit reasoning step, so training pipelines receive no pressure to ground chain-of-thought in what the model *sees*;Jiang et al. (2025b); Zhang et al. (2024); Shao et al. (2024) RL post-training therefore optimizes conversational fluency while silently tolerating physical implausibility. The result is **a generation of MLLMs that can *describe* images eloquently yet still misjudge which way a car is moving**.

To close this evaluation gap, we introduce **MVPBench**, a Multi-path Visual Physics benchmark that turns the spotlight on vision-centric reasoning. MVPBench contains 1,211 carefully curated examples across three real-world domains: *i.* hands-on physics experiments (electromagnetic induction, heat conduction, collisions), *ii.* exam-style word problems requiring symbolic or commonsense reasoning, and *iii.* spatial-transformation tasks that challenge 3D understanding (viewpoint shifts, object rearrangement). Each example pairs *multi-image evidence* with *multiple valid CoT paths*, forcing models to justify every step in view of changing visuals. To evaluate such rich annotations, we introduce a **graph-based CoT metric suite** that represents each reasoning chain as a directed acyclic graph of atomic facts and then assesses step-wise fidelity through exact or fuzzy graph matching, measures text–image grounding with automated alignment scores, and quantifies multi-path coverage by rewarding diverse yet logically valid reasoning flows. MVPBench thus re-aligns the compass: genuine physical understanding demands that models *see, think, and prove*—not merely narrate.

Extensive experiments reveal two key insights: *i.* Providing models with the full image sequence boosts performance by up to 21% points-evidence that temporal context matters. *ii.* Contrary to conventional wisdom, RL-based post-training *reduces* visual-physics scores on MVPBench by 2% points, indicating that current reward designs sacrifice grounded reasoning for coherence.

**To summarize, this paper makes the following contributions:***i.* To the best of our knowledge, **MVPBench** is the first benchmark to combine real-world visual physics, multi-image inputs, and *multi-path* CoT annotations. *ii.* A **graph-based evaluation toolkit** that jointly measures reasoning fidelity, visual grounding, and path diversity. *iii.* The first comprehensive study showing that widely adopted RL alignment can impair spatial reasoning, calling for vision-centric reward design.

## 2 RELATED WORKS

**Limitations of Multi-modal Large Language Models in Visual Physical Reasoning.** Despite rapid progress, recent studies show that MLLMs still exhibit substantial weaknesses in understanding the physical world from visual input. Their visual physical understanding remains fragile Liu et al. (2024); Guo et al. (2024); Bonnen et al. (2024), and they face major challenges when reasoning over

visual perception in complex scenes Zhang et al. (2025b); Zheng et al. (2025a); Bi et al. (2025). In terms of physical discipline knowledge, models show limited ability to perform multimodal reasoning over discipline-level problems He et al. (2024). When tasked with predicting physical interactions and long-term object dynamics, they often fail to capture the underlying causal structure Yi et al. (2020); Bear et al. (2022) Moreover, MLLMs struggle to accurately infer object properties and latent states in physics-based scene evaluations Wang et al. (2023b); Balazadeh et al. (2025). Even their spatial reasoning, while improving, frequently breaks down on visual tasks that require precise understanding of spatial relations and configurations Chen et al. (2024a). Taken together, these findings highlight the need for more comprehensive and rigorous benchmarks that specifically target the visual physical and spatial reasoning capabilities of MLLMs.

**Physical Comprehension Datasets.** These datasets have become a crucial area of focus, posing a significant challenge for MLLMs. Early physical benchmarksBear et al. (2022); Zhu et al. (2023); Tung et al. (2023)were developed around simple physical scene reasoning. Inspired by research on infant intuitive physics, the studyRiochet et al. (2020) evaluate innate understanding of models in the physical world. In other aspects of physical datasets, existing benchmarksHe et al. (2024); Jiang et al. (2024); Lu et al. (2022); Hao et al. (2025); Zhang et al. (2025c) to evaluate physics problems mainly focus on commonsense reasoning based on language knowledge. Spatial benchmarksWang et al. (2023a); Yang et al. (2024); Shiri et al. (2024); Li et al. (2024), on the other hand, emphasize spatial perception and reasoning in 3D scenes, illustrating the early stages of world model. Recent effortChow et al. (2025) has expanded to comprehensively assess understanding of models in physical scenes across various tasks, though they still fail to fully encompass real-world physical knowledge. **By introducing visual CoT as inputs, it forces models to reason across images, making it a closer approximation to the analysis of complex physical scenes in the real world.**

**Table 1** **Comparison of MVPBench with existing benchmarks for physical understanding.** MVPBench covers a broader range of physical reasoning categories, supports multi-perspective chain-of-thought evaluation, and provides CoT annotations. In the data format, TC indicates that the dataset utilizes textual CoT, VC means the use of visual CoT as input, and Vc signifies all that the data is constructed in a vision-centric manner.

| Benchmark | Data category | | | | CoT Evaluation | | | Data format | | |
|---|---|---|---|---|---|---|---|---|---|---|
| | Physics experiments | Physics problems | Spatial relations | Dynamic prediction | Quality | Diversity | Efficiency | Vc | TC | VC |
| PhysBenchChow et al. (2025) | | | ✓ | ✓ | | | | | | ✓ |
| PhysionBear et al. (2022) | | | | ✓ | | | | | | |
| PhysReasonZhang et al. (2025c) | ✓ | | | | ✓ | | | | ✓ | |
| PhysGameCao et al. (2024) | | | | ✓ | | | | ✓ | | |
| ContPhyZheng et al. (2024) | | | | ✓ | | | | | | |
| EmbSpatialDu et al. (2024) | | | ✓ | | | | | | | |
| **MVPBench** | ✓ | ✓ | ✓ | ✓ | ✓ | ✓ | ✓ | ✓ | ✓ | ✓ |

## 3 MVPBench

The motivation for constructing the MVPBench benchmark stems from recognizing significant gaps in the current capability of MLLM to deeply comprehend and reason about the physical world. Existing benchmarks emphasize isolated aspects such as static scene understanding, physics-based reasoning, or basic spatial awareness, leaving unaddressed the integration of physical reasoning with complex visual inputs. Therefore, MVPBench aims to rigorously evaluate abilities of MLLMs to visually reason about diverse physical phenomena in scenarios closely resembling real-world complexities.

To ensure comprehensive coverage of visual reasoning skills, MVPBench incorporates carefully curated data across multiple distinct yet complementary domains: 1) *Physics Experiments* tests the understanding of sequential physical processes through multi-step visual inference. 2) *Physics Problems* challenges models to interpret advanced, visually grounded physics questions from academic examinations. 3) *Spatial Relations* assesses spatial perception judgment across various scenarios. 4) *Dynamic Prediction* evaluates the predictive capabilities of models regarding dynamically evolving physical interactions. Collectively, these diverse yet targeted subdomains ensure MVPBench not only addresses existing evaluation gaps but also significantly extends the reasoning depth, robustness, and versatility of models. Details of data analysis are provided in Appendix C.

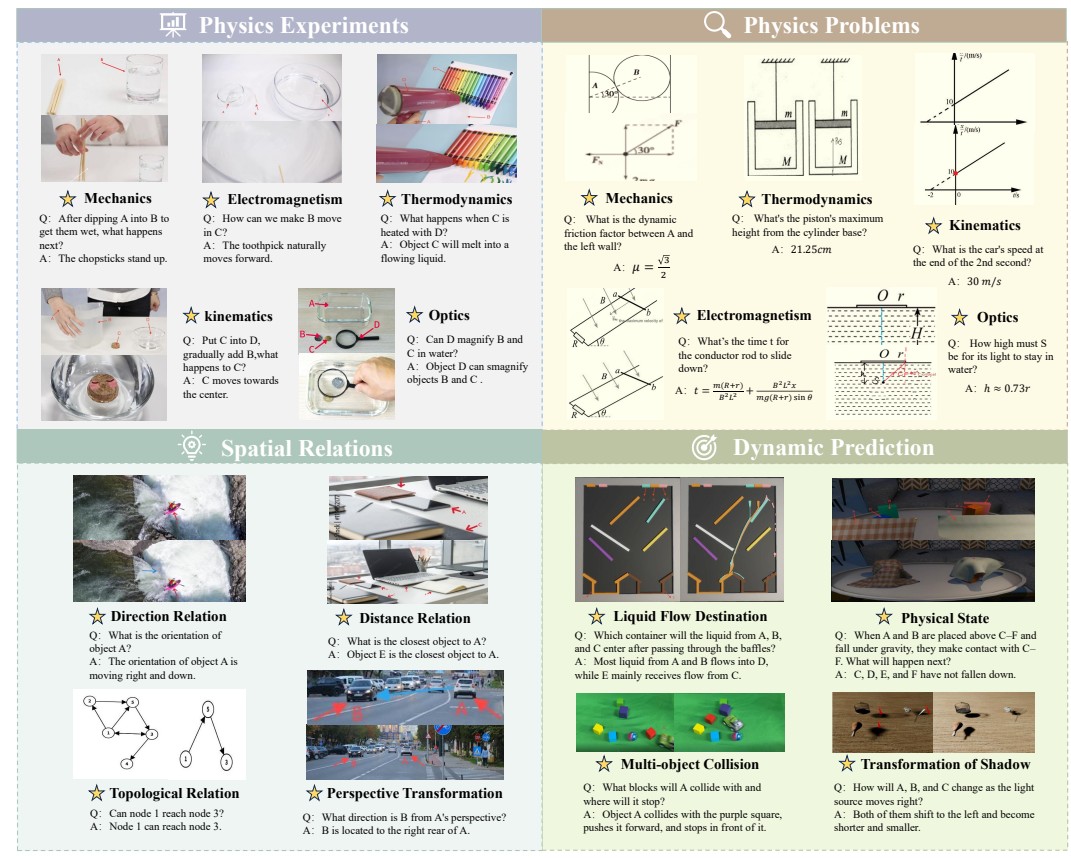

**Figure 2 Examples from MVPBench across four categories.** Each example includes an initial scene followed by reasoning steps. Target objects are marked with red arrows and labeled with letters to reduce textual bias.

## 3.1 DATA GENERATION

**Physics Experiments.** We scraped publicly available physics experiment videos, manually filtered them, and archived the curated clips as MP4 files. From each video, we extracted key frames depicting (i) the initial setup, (ii) critical intermediate steps, and (iii) the final results. Salient objects were highlighted with arrows while all textual cues were omitted, forcing models to infer solely from visual cues, with GPT-4 generating the corresponding scene descriptions. The intermediate steps encompass essential logical reasoning processes required to complete each experiment. ***To evaluate multi-path reasoning verification capability of MLLMs, recorded multiple chains of thought for each instance.*** All assets are stored in a structured JSON schema that includes mechanics, thermodynamics, electromagnetism, optics and kinematics. The remaining subsets empoly the same format as the JSON detailed above, and we omit related discussion in the following section.

**Physics Problems.** On one hand, we crawled and manually filtered all the problems from relevant websites, compiling them into PDF files, which were then converted into Markdown format via OCR and manually aligned. On the other hand, the data was with examples from the PhysReason-miniZhang et al. (2025c) dataset. All problems are tightly coupled to images and drawn from examinations in several countries (predominantly Chinese college entrance examination) for their open-ended formats that demand advanced reasoning. After meticulous verification, we extracted key reasoning steps and final answers. These steps include both textual and visual components, with the image segment forming an additional input alongside the original image. The questions cover five subcategories including mechanics, thermodynamics, electromagnetism, optics, and kinematics.

**Spatial Relations.** Spatial relation reasoning is a crucial area in understanding of the physical world. To address this gap, we have pre-designed four main subcategories to evaluate perception of spatial relations: **(1) Direction judgment**: This subcategory formulates problems concerning the directional judgment of various objects. **(2) Distance estimation**: This subset encompasses problems related to

estimating the distance relation of different objects. **(3) First view transformation**: This subcategory addresses issues pertaining to direction judgment from a egocentric viewpoint regarding various objects. **(4) Topological relation judgment**: This subcategory focuses on problems associated with reachability within directed graphs. The first three subcategories manually screened original images from public websites, and the fourth subcategory constructed images using the Graph Editor tool.

**Dynamic Prediction.** To investigate whether MLLMs can predict time-varying physical outcomes through visual reasoning, we introduce a Dynamic Prediction subset comprising four subcategories: **Multi-object Collision, Liquid Diversion, Physical state and Shadow Transformation** predict. This subset utilizes the PhysBench Chow et al. (2025) benchmark, which provides high-quality dynamic scene videos. All samples are adapted and extended from PhysBench to ensure high-quality video frames. For each sample, we extract multiple temporally spaced key frames from the corresponding video to form multi-image inputs, annotating salient objects with arrows.

# 4 CoT Evaluation Method

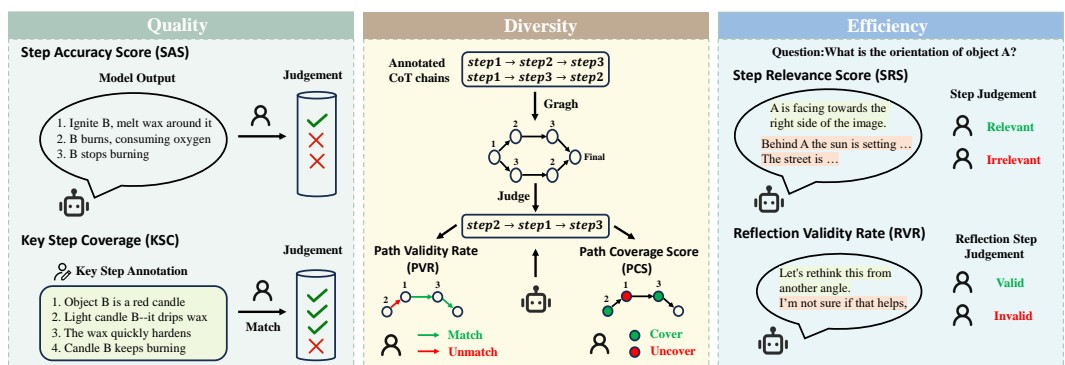

**Figure 3 Evaluation framework for multi-path Chain-of-Thought (CoT) reasoning.** MVPBench introduces a comprehensive protocol to evaluate CoT reasoning from three perspectives: quality, diversity, and efficiency. **For CoT diversity, we propose a graph-based multi-path evaluation method** that quantifies the ability of a model to explore alternative reasoning routes via Path Validity Rate (PVR) and Path Coverage Score (PCS), advancing beyond prior single-path metrics.

Existing CoT evaluation methods often simplify reasoning assessment to a binary judgment of the final answer, overlooking the internal reasoning steps. To address this limitation, we propose a holistic CoT evaluation suite that captures the reasoning process across multiple dimensions, offering a finer-grained understanding of reasoning capabilities of MLLMs. *Notably, we are the first to introduce an evaluation metric for assessing multi-path reasoning ability of models, which complements traditional correctness and reflection assessments.* Details are presented in Section 4.1 (correctness), Section 4.2 (multi-path reasoning), and Section 4.3 (reflection quality).

## 4.1 CoT Quality Evaluation

To evaluate the correctness of CoT reasoning, we extend existing interpretable metrics by incorporating both step-wise accuracy and final answer correctness. While prior work such as Jiang et al. (2025b) focused on intermediate informativeness, they overlook the contribution of the final answer to overall quality. Inspired by Zhang et al. (2024), we introduce a **weighted scoring framework** that balances the quality of intermediate steps with the correctness of the final prediction.

**Step Accuracy Score (SAS).** We prompt GPT-4o OpenAI (2024) to decompose each CoT prediction into steps, categorized as logical inference, image captioning, or background/numerical computation (depending on the dataset). Each step is binary-judged for correctness based on alignment with references or logical/visual validity. SAS is computed as the proportion of correct steps.

**CoT Reasoning Score (CRS).** To combine step-wise correctness and final answer validity, we define a weighted reasoning score as $\text{CRS} = \alpha \cdot \text{SAS} + (1 - \alpha) \cdot \text{Correct}(s_A)$, where $\text{Correct}(s_A) \in \{0, 1\}$ denotes whether the final answer is correct, and $\alpha$ is set to 0.7 by default.

**Key Step Coverage (KSC).** We also measure the proportion of annotated key reasoning steps that appear in the model output, serving as a recall-style indicator of reasoning completeness.

## 4.2 CoT Diversity Evaluation

While some recent studies have acknowledged the need for multi-path reasoning evaluation, significant gaps remain. Zhang et al. (2024) emphasizes that rigid ground-truth templates fail to capture the diversity of reasoning styles, calling for adaptive key-step extraction. Similarly, Jiang et al. (2025b) and Chow et al. (2025) annotate multiple reasoning paths but lack systematic metrics to measure the ability of models to generate and validate diverse CoT trajectories.

Accordingly, we introduce **CoT Diversity Evaluation (CDE)**, a graph-based framework for assessing the ability of models to generate logically valid and distinct reasoning chains, with three key stages:

- **Reference Graph Construction.** Each annotated instance is converted into directed graphs, with key steps as nodes and logical flows as edges.
- **Model Path Embedding.** We map the model-generated reasoning steps into the reference graph by parsing them into directed edge sequences.
- **Path Matching and Metric Computation.** We define three core metrics for multi-path evaluation:
  - **Path Validity Rate (PVR)**: Proportion of model edges matching the reference graph.
  - **Path Coverage Score (PCS)**: Normalized length of the longest matched sub-path.

**Path Count Adjustment.** To fairly compare models with differing numbers of generated and reference paths, we define adjusted versions of the above metrics.

Let $N_p$ and $N_{gt}$ denote the numbers of predicted and reference paths, respectively. The adjusted path validity rate is defined as Path Validity Rate$_{\text{adj}}$ = PVR $\times \frac{\min(N_p, N_{gt})}{N_{gt}}$, and the adjusted path coverage score is given by Path Coverage Score$_{\text{adj}}$ = PCS $\times \exp\left(-\alpha \cdot \left(\frac{N_p}{N_{gt}} - 1\right)\right)$, where $\alpha$ controls the penalty for over-generation: higher values enforce stricter adherence to the reference count, while lower values allow more flexibility.

**Structure-Tolerant Matching.** From our preliminary experiments we observed that DAG-based matching is overly sensitive to small structural variations: logically equivalent reasoning paths may still be penalized if node or edge order differs. To address this, we introduce a **Graph Edit Distance (GED)** similarity, which measures the minimal number of edit operations (insertions, deletions, substitutions) needed to transform the model graph into the reference graph. We map this distance into a smooth similarity score as Sim $= \exp(-\gamma \cdot \text{GED})$, where $\gamma = 0.5$ controls sensitivity to structural differences. We then define the CoT Match Score (CMS) by combining path-level validity and coverage with structure tolerance: CMS $= \lambda \cdot \frac{(\text{PVR+PCS})}{2} + (1 - \lambda) \cdot \text{Sim}$, where $\lambda = 0.7$ balances diversity against robustness to structural variations. This adjustment enables CDE to more faithfully evaluate both the logical validity and the structural flexibility of model-generated reasoning paths.

## 4.3 CoT Efficiency Evaluation

The efficiency of reasoning is also crucial for evaluating CoT quality. Models like o1 generate excessively long reasoning chains with extensive reflection and verification steps. To capture this aspect, we evaluate the relevance of reasoning steps and the validity of reflective ones.

**Step Relevance Score (SRS).** While long reasoning sequences enable deeper analysis, they often include irrelevant descriptions unrelated to solving the task. We partition the model's reasoning into steps and instruct GPT-4o to identify all relevant steps $P_{\text{relevant}}$. A step is considered relevant if its major content directly contributes to problem-solving. SRS, similar to SCS, is defined as the proportion of relevant steps among all generated steps.

**Reflection Validity Rate (RVR).** Reflective reasoning can strengthen CoT performance by identifying errors or providing additional justification, but not all reflections are helpful—some may be redundant or incorrect. We define a reflection step as valid if it (i) identifies a previous error or (ii) offers new supporting reasoning. Reflection quality is then measured as the proportion of valid reflections $R_{\text{valid}}$, detected through linguistic cues such as "Wait" or "Alternatively".

**Table 2  CoT reasoning performance on MVPBench across three dimensions.** We assess open- and closed-source MLLMs on *CoT Quality* (SAS, KSC, CRS), *CoT Diversity* (PVR, PCS, CMS), and *CoT Efficiency* (SRS, RVR, Avg), under *Single* and *Multi* image settings. Best single-image results and largest multi-image gains are highlighted for closed-source and open-source models. ↑ indicates performance improvement with multi-image input, ↓ indicates a drop, and ∗ denotes invalid outputs. Additional evaluation results for closed-source models and human performance benchmarks are presented in the Appendices A.1 and B.1, respectively.

| Model | CoT Quality | | | | | | CoT Diversity | | | | | | CoT Efficiency | | | | | |
|---|---|---|---|---|---|---|---|---|---|---|---|---|---|---|---|---|---|---|
| | SAS | | KSC | | CRS | | PVR | | PCS | | CMS | | SRS | | RVR | | Avg | |
| | Single | Multi | Single | Multi | Single | Multi | Single | Multi | Single | Multi | Single | Multi | Single | Multi | Single | Multi | Single | Multi |
| *Open-source MLLMs* | | | | | | | | | | | | | | | | | | |
| LLaVA-OV-72B Li et al. (2025a) | 53.09 | * | 29.47 | * | 36.49 | * | 63.44 | * | 70.00 | * | 74.01 | * | 96.91 | * | 99.55 | * | 98.23 | * |
| LLaVA-CoT Xu et al. (2024) | 48.47 | 8.58↑ | 30.21 | 2.23↑ | 32.58 | 9.01↑ | 28.87 | 10.32↑ | 51.89 | 3.75↑ | 48.43 | 9.02↑ | 97.63 | 0.49↓ | 99.64 | 0.12↓ | 98.64 | 0.49↓ |
| InternVL2.5-78B Chen et al. (2024b) | 56.35 | 10.12↑ | 42.42 | 5.45↑ | 43.98 | 4.45↑ | 67.28 | 8.43↑ | 72.09 | 4.79↑ | 70.82 | 5.12↑ | 96.89 | 0.83↓ | 99.45 | 0.50↑ | 98.17 | 0.16↓ |
| InternVL2.5-78B-MPO Wang et al. (2024b) | 55.77 | 7.80↑ | 41.87 | 5.63↑ | 43.76 | 8.51↑ | 72.80 | 9.34↑ | 76.08 | 5.61↑ | 79.33 | 8.11↑ | 97.88 | 1.67↓ | 99.32 | 0.28↓ | 98.60 | 0.98↓ |
| InternVL3-78B Zhu et al. (2025) | 57.80 | 9.26↑ | 46.20 | 5.49↑ | 47.48 | 9.25↑ | 66.06 | 7.02↑ | 70.61 | 7.53↑ | 76.77 | 8.65↑ | 97.54 | 0.35↑ | 99.52 | 0.11↓ | 98.53 | 0.13↑ |
| InternVL3-78B-Instruct Zhu et al. (2025) | 55.86 | 9.53↑ | 42.15 | 3.51↑ | 44.24 | 8.63↑ | 68.41 | 9.78↑ | 72.41 | 3.41↑ | 76.23 | 8.38↑ | 96.88 | 0.29↑ | 99.92 | 0.50↑ | 98.40 | 0.10↓ |
| Qwen2.5-VL-7B Bai et al. (2025) | 52.40 | 3.11↑ | 36.54 | 1.73↑ | 39.24 | 4.32↑ | 64.43 | 5.83↑ | 73.70 | 2.12↑ | 74.00 | 3.86↑ | 93.59 | 0.30↑ | 99.26 | 0.02↓ | 96.43 | 0.14↑ |
| Qwen2.5-VL-72B Bai et al. (2025) | 57.15 | 5.55↑ | 43.29 | 5.33↑ | 46.08 | 7.24↑ | 74.73 | 6.76↑ | 78.97 | 6.12↑ | 82.43 | 7.34↑ | 97.46 | 1.50↓ | 99.43 | 0.24↑ | 98.45 | 0.63↓ |
| QVQ-72B Qwen Team (2024) | 68.28 | 2.49↑ | 44.63 | 0.76↓ | 53.83 | 0.88↓ | * | * | * | * | * | * | 85.29 | 3.82↑ | 56.27 | 3.04↑ | 70.93 | 3.28↑ |
| *Closed-source MLLMs* | | | | | | | | | | | | | | | | | | |
| GPT-4o OpenAI (2024) | 63.26 | 20.30↑ | 46.39 | 14.75↑ | 50.45 | 21.41↑ | 68.04 | 13.22↑ | 72.38 | 10.01↑ | 81.34 | 13.04↑ | 98.42 | 1.26↓ | 99.39 | 0.28↑ | 98.90 | 0.49↓ |
| OpenAI o3 OpenAI (2025) | 75.29 | 15.87↑ | 50.64 | 11.52↑ | 59.11 | 15.83↑ | 68.85 | 9.81↑ | 74.91 | 10.24↑ | 76.65 | 9.97↑ | 99.43 | 2.31↓ | 99.52 | 0.13↓ | 99.48 | 1.09↓ |
| Claude 3.7 Sonnet Anthropic (2025) | 64.41 | 16.12↑ | 45.66 | 11.95↑ | 50.87 | 15.22↑ | 73.70 | 12.81↑ | 75.79 | 12.04↑ | 79.08 | 13.38↑ | 97.76 | 0.13↑ | 97.34 | 2.23↑ | 97.55 | 1.18↑ |

## 5  COMPREHENSIVE EVALUATION OF CoT-BASED MULTIMODAL REASONING

**Overall Results.** Table 2 reports model performance across three CoT evaluation dimensions using SAS, KSC, and SRS for both logical inference and image captioning. Diversity is assessed via PVR and RCS, and robustness is measured by averaging SRS and RVR, with RVR set to 100 for models lacking reflection ability. Table 3 complements this by presenting subcategory-level evaluation across all CoT metrics on MVPBench. Model and setup details are in Appendix H.

GPT-4o demonstrates strong overall performance, while OpenAI o3 surpasses it in quality and efficiency, achieving the highest scores. Among open-source models, the InternVL series is most competitive, with InternVL3-vl-78B and MPO-tuned InternVL2.5 showing strong performance across all dimensions. QVQ performs well in CoT quality but lacks robustness, often producing verbose and loosely related content, from which we derive the following key observations.

***CoT Diversity Does Not Guarantee High Reasoning Accuracy.*** While diversity helps explore multiple reasoning paths, our results show it does not inherently improve reasoning quality. For example, Qwen2.5-VL-72B achieves the highest diversity but underperforms QVQ-72B in quality, despite the latter lacking diversity evaluation. This suggests a trade-off: greater diversity may lead to less focused or accurate reasoning if not properly guided. In contrast, OpenAI o3 attains top quality with moderate diversity, highlighting the importance of goal-directed reasoning.

***Reflection enhances quality but with limited reliability.*** As shown in Table 2, QVQ with reflection surpasses its base model Qwen2.5-VL-72B by 7.75% in CRS and 11.13% in SAS, even with longer CoT sequences, approaching GPT-4o in quality. However, its reflection validity rate is only about 56%, meaning nearly half of reflection attempts fail to aid accuracy, which compromises efficiency and introduces redundant reasoning steps.

***Long CoT Models May Be More Prone to Distraction.*** Models generating longer CoT tend to exhibit lower relevance, often producing content unrelated to the question, reflected by lower KSC scores (compared to QVQ). Some short-CoT models like LLaVA-OV-72B also show low relevance, usually due to repetitive outputs on specific question types. Fine-grained analysis shows models often lose focus when describing images, generating exhaustive but irrelevant captions.

***Post-training may harm generalization.*** While post-training—particularly mixed preference optimization (MPO)—is frequently employed to align models more closely with specific downstream tasks, it does not universally enhance CoT reasoning quality. As in Figure 5, InternVL2.5-78B-MPO underperforms its base counterpart InternVL2.5-78B, and InternVL3-78B similarly trails InternVL3-78B-Instruct in Physics Experiments subset. Although MPO can effectively boost performance on human-preference-aligned subsets such as physics questions, it tends to negatively impact subsets requiring stronger visual perception or temporal prediction capabilities. This phenomenon suggests that

MPO may introduce distributional biases or lead to overfitting to specific tasks, thereby compromising generalization, visual grounding, and multimodal coherence—particularly evident in visual-centric reasoning tasks. MVPBench, with its comprehensive and balanced design across multiple reasoning categories, effectively highlights these limitations.

# 6 UNDERSTANDING THE EVALUATIVE POWER OF MVPBENCH

**Table 3 Subcategory-level evaluation of CoT reasoning in MVPBench.** We present subcategory-level scores for three core reasoning metrics and evaluated across both open- and closed-source MLLMs. Top-performing models within each category are highlighted in blue (open-source) and red (closed-source). For models (SpaceQwen2.5 and SpaceThinker) fine-tuned specifically for spatial reasoning and their corresponding base model (Qwen2.5VL-3B), we evaluate only on the Spatial-Relation subset, * denotes no output.

| Model | Phys-Experiment | | | Phys-Problems | | | Spatial-Relation | | | Dyn-Prediction | | |
|---|---|---|---|---|---|---|---|---|---|---|---|---|
| | Quality | Diversity | Efficiency | Quality | Diversity | Efficiency | Quality | Diversity | Efficiency | Quality | Diversity | Efficiency |
| *Open-source MLLMs* | | | | | | | | | | | | |
| LLaVA-OV-72B Li et al. (2025a) | 37.21 | 66.61 | 94.77 | 32.94 | 79.72 | 99.05 | 34.16 | 59.79 | 99.36 | 41.66 | 89.93 | 99.72 |
| LLaVA-CoT Xu et al. (2024) | 33.79 | 52.34 | 97.35 | 20.86 | 45.46 | 98.97 | 31.89 | 54.31 | 98.45 | 43.77 | 41.61 | 99.78 |
| InternVL2.5-78B Chen et al. (2024b) | 43.95 | 73.38 | 94.25 | 47.44 | 71.32 | 98.83 | 39.75 | 71.08 | 99.59 | 44.78 | 87.49 | 100 |
| InternVL2.5-78B-MPO Wang et al. (2024b) | 41.60 | 79.43 | 97.19 | 51.54 | 75.87 | 98.97 | 37.83 | 71.42 | 98.48 | 44.06 | 90.60 | 99.76 |
| InternVL3-78B Zhu et al. (2025) | 37.00 | 83.39 | 91.49 | 58.26 | 68.23 | 98.92 | 39.31 | 70.43 | 99.14 | 46.68 | 88.05 | 99.95 |
| InternVL3-78B-Instruct Zhu et al. (2025) | 42.01 | 74.57 | 94.87 | 52.64 | 69.79 | 99.81 | 38.10 | 70.96 | 98.96 | 44.20 | 89.58 | 99.96 |
| Qwen2.5-VL-7B Bai et al. (2025) | 37.00 | 80.15 | 91.49 | 42.34 | 67.10 | 98.55 | 35.20 | 68.57 | 95.82 | 40.30 | 82.16 | 99.85 |
| Qwen2.5-VL-72B Bai et al. (2025) | 41.19 | 82.59 | 96.72 | 57.01 | 79.69 | 99.36 | 39.18 | 59.67 | 98.06 | 46.94 | 98.75 | 99.65 |
| QVQ-72B Qwen Team (2024) | 49.63 | 0.00 | 71.65 | 60.97 | 0.00 | 63.71 | 38.50 | 0.00 | 69.24 | 66.20 | 0.00 | 79.13 |
| Qwen2.5VL-3BBai et al. (2025) | * | * | * | * | * | * | 22.24 | 13.40 | 95.03 | * | * | * |
| SpaceQwen2.5-VL-3BJia et al. (2025) | * | * | * | * | * | * | 20.84 | 34.86 | 93.72 | * | * | * |
| SpaceThinker-Qwen2.5VL-3B Chen et al. (2025) | * | * | * | * | * | * | 23.93 | 31.15 | 97.87 | * | * | * |
| *Closed-source MLLMs* | | | | | | | | | | | | |
| GPT-4o OpenAI (2024) | 50.21 | 76.36 | 97.53 | 52.29 | 69.52 | 98.77 | 43.64 | 69.33 | 99.72 | 52.35 | 91.14 | 99.59 |
| OpenAI o3 OpenAI (2025) | 57.73 | 75.58 | 97.44 | 65.36 | 68.57 | 99.06 | 43.92 | 71.26 | 99.83 | 69.44 | 91.18 | 99.71 |
| Claude 3.7 Sonnet Anthropic (2025) | 49.13 | 78.97 | 97.38 | 57.02 | 72.15 | 94.71 | 42.41 | 72.71 | 99.67 | 54.92 | 92.47 | 98.45 |

Our dataset, MVPBench, is specifically constructed to test multimodal reasoning under diverse and fine-grained physical scenarios. We explore its impact on evaluation outcomes from there perspectives: the effectiveness of fine-tuning spatial reasoning, category diversity and input modality.

***The effectiveness of specialized fine-tuning strategies aimed explicitly at spatial reasoning.*** To further explore MLLMs specifically fine-tuned for spatial reasoning capabilities, we selected three representative models for rigorous comparison: Qwen2.5VL-3BBai et al. (2025) as a baseline model without specialized spatial reasoning fine-tuning, and two models (SpaceQwen2.5-VL-3BJia et al. (2025) and SpaceThinker-Qwen2.5VL-3BChen et al. (2025)) employing different specialized fine-tuning strategies to enhance spatial reasoning. We conducted rigorous evaluations on the Spatial-Relation subset within MVPBench, comparing the models across three dimensions: CoT Quality, Diversity, and Efficiency. The detailed results are presented in the table 3. Compared with the baseline Qwen2.5VL-3B, both spatially fine-tuned models show smaller CoT-Quality drops on multi-image tasks. SpaceThinker-Qwen2.5VL-3B beats Qwen2.5VL-3B in CoT-Quality, indicating that synthetic reasoning-trace fine-tuning strengthens multi-step visual reasoning. CoT-Diversity rises markedly with fine-tuning (34.86% and 31.15%) versus the baseline's 13.40%, yielding richer, more flexible reasoning paths. The baseline gains a bit in CoT-Efficiency but loses CoT-Quality under multi-image complexity, while SpaceThinker-Qwen2.5VL-3B achieves the highest efficiency (97.87%) with a slight dip for multi-image input. Overall, targeted spatial-reasoning fine-tuning delivers clear gains in quality and diversity that outweigh minor efficiency trade-offs.

***Category diversity influences evaluation difficulty.*** MVPBench spans a variety of physical reasoning subcategories, each posing distinct challenges. We observe that model performance varies substantially across these categories, underscoring the impact of task type on evaluation difficulty. For example, InternVL3-78B achieves a Quality score of 58.26 on the more abstract *Phys-Problems* category, but performs better with a score of 66.68 on the more concrete *Dyn-Prediction* tasks (see Table 3). Notably, across all open-source models, the *Spatial-Relation* subset yields the lowest average Quality score (37.10), suggesting it poses the greatest challenge. This indicates that current MLLMs still struggle with fine-grained spatial reasoning, revealing a critical gap in their perceptual and relational understanding of physical scenes. This performance gap illustrates how reasoning

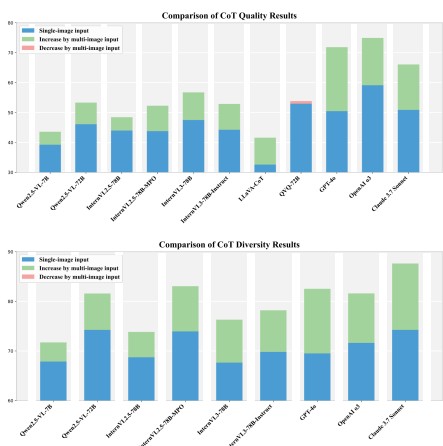

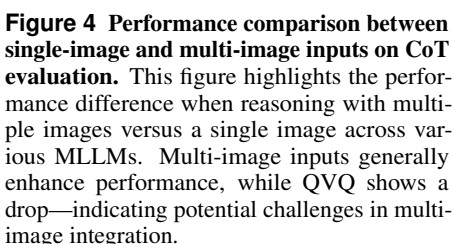

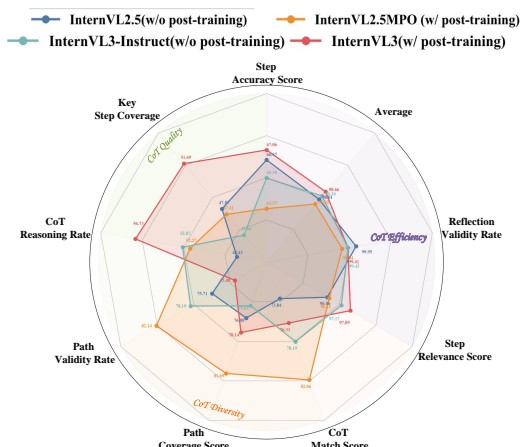

**Figure 4  Performance comparison between single-image and multi-image inputs on CoT evaluation.** This figure highlights the performance difference when reasoning with multiple images versus a single image across various MLLMs. Multi-image inputs generally enhance performance, while QVQ shows a drop—indicating potential challenges in multi-image integration.

**Figure 5  CoT Performance of MLLMs with post-training versus without post-training.** InternVL2.5 and InternVL3-instruct represent models without post-training, whereas InternVL2.5-MPO and InternVL3 denote their post-trained counterparts. Please note that each metric axis has its own independent scale. The results clearly indicate that post-training often fails to enhance the reasoning performance of models and degrades it.

complexity varies by category and highlights the importance of category-aware evaluation for robust and meaningful model comparisons.

***Multi-image input significantly boosts model performance.*** To evaluate the impact of input modality, we conducted comparative experiments using both single-image and multi-image inputs under identical prompts and evaluation metrics. This design isolates the effect of visual input quantity, allowing for a controlled analysis of performance variance. As illustrated in Fig 4, nearly all models benefit from multi-image inputs, achieving notable gains in both CoT Quality and Diversity scores. Closed-source models show particularly striking improvements, with GPT-4o leading the trend—its CoT Quality score rises from 50 to 72, a relative increase of 44%, and its Diversity score jumps from 70 to 85, a 21% improvement. Other closed-source models like Claude 3.7 Sonnet and OpenAI o3 also exhibit significant gains, with Quality scores increasing by 15% and Diversity by 13%. Open-source models, such as InternVL3-78B, show more modest improvements, rising from a Quality score of 47.5 to 56.7 (a 19% increase) and a Diversity score improvement of around 10%. However, QVQ-72B is an outlier, showing a performance drop of roughly 1-2 points in quality, indicating potential challenges in multi-image integration. Overall, these results highlight the superior adaptability of closed-source models, particularly GPT-4o, in leveraging multi-image inputs to enhance fine-grained physical reasoning and diversity in responses.

## 7  CONCLUSION

We introduce MVPBench, a benchmark designed to rigorously evaluate visual chain-of-thought reasoning in multimodal large language models (MLLMs). It target tasks that require grounded, multi-step inference over visual evidence and goes beyond surface-level image description. Our evaluation reveals than even state-of-the-art models like GPT-4o and OpenAI o3 often struggle with physical reasoning. To diagnose these failures, we introduce a graph-based CoT consistency metric to assess reasoning validity, uncovering frequent violations of basic physical principles. Notably, we find that reinforcement learning-based alignment can impair physical reasoning, highlighting a misalignment between current fine-tuning strategies and the demands of physical perceptual reasoning. These findings call for post-training strategies that better integrate visual grounding, causal structure, and structured explananation in MLLMs.

## 8 ETHICS STATEMENT

This work does not involve human subjects, personally identifiable information, or sensitive data. All datasets used are publicly available and released under appropriate licenses. The research was conducted in compliance with ethical standards, with no foreseeable risks of harm, privacy violations, or misuse.

## 9 REPRODUCIBILITY STATEMENT

We provide a comprehensive description of the dataset collection and preprocessing steps in the appendix C, including detailed documentation to ensure clarity and transparency. The implementation details and evaluation settings for each benchmarked model are also thoroughly reported in the appendix H. To further promote reproducibility, we have included all the code, configuration files and experimental scripts in the supplementary materials, and provided the access link to our dataset at the appendix overview.

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

APPENDIX OVERVIEW

Our supplementary includes the following sections:

- **Section A: More experiment results.** Extended Empirical Analysis on Closed-source and Post-trained Models.
- **Section B: More Exploration.** Analysis of human performance and error analysis.
- **Section C: More Dataset Details.**
- **Section D: More Qualitative Examples.** More visualization of our evaluation demos.
- **Section E: Limitations.** Discussion of limitations of our work.
- **Section F: Broader impacts.** Discussion of societal impacts of our work.
- **Section G: Detailed Evaluation prompts.**
- **Section H: Setup.** Details for model design, implementation.
- **Section I: The Use of LLMs.**
- **Section J: Rebuttal.**

To further promote reproducibility, we provide our dataset, which can be accessed via an anonymous link.

# A  MORE EXPERIMENT RESULTS

## A.1  MORE CLOSED-SOURCE MODEL EXPERIMENTS

**Table 4 Additional Evaluation Results for Closed-Source Models on CoT Reasoning Performance across Three Dimensions in MVPBench.** ↑ indicates performance improvement with multi-image input, ↓ indicates a drop.

| Model | CoT Quality | | | | | | CoT Diversity | | | | | | CoT Efficiency | | | | | |
|---|---|---|---|---|---|---|---|---|---|---|---|---|---|---|---|---|---|---|
| | SAS | | KSC | | CRS | | PVR | | PCS | | CMS | | SRS | | RVR | | Avg | |
| | Single | Multi | Single | Multi | Single | Multi | Single | Multi | Single | Multi | Single | Multi | Single | Multi | Single | Multi | Single | Multi |
| *Closed-source MLLMs* | | | | | | | | | | | | | | | | | | |
| Gemini-2.5-flash-preview-04-17 Deepmind (2024) | 60.56 | 12.32↑ | 49.29 | 8.54↑ | 50.05 | 11.63↑ | 56.44 | 9.23↑ | 59.35 | 7.12↑ | 57.20 | 8.45↑ | 97.59 | 0.37↓ | 92.00 | 2.00↑ | 94.71 | 0.82↑ |
| Grok3xAI (2025) | 62.48 | 3.44↑ | 52.05 | 4.13↑ | 52.69 | 4.50↑ | 61.57 | 10.13↑ | 68.05 | 6.89↑ | 63.78 | 8.43↑ | 89.55 | 2.53↓ | 86.00 | 6.26↑ | 87.77 | 1.87↑ |

To evaluate additional closed-source models, we randomly sampled 25 instances from each sub-dataset of MVPBench, resulting in 100 samples in total. As shown in Table 4 and Table 11, the results of these models largely confirm the trends observed with tested models discussed earlier: performance varies notably across different sub-datasets, and multi-image input consistently leads to substantial improvements. Interestingly, Gemini Deepmind (2024) demonstrates strong quality in the *Physics Experiments* subset, yet performs surprisingly poorly in the *Spatial Relations* task—even falling behind several open-source models.

## A.2  MORE POST-TRAINING MODEL EXPERIMENTS

To further investigate the impact of post-training on model generalization, we conducted additional experiments comparing different base models and distinct post-training methods. Specifically, we compared two base models without post-training, Qwen2.5VL-7B and Qwen2VL-2B, against their respective post-trained counterparts: MM Eureka-7B, which employs large-scale rule-based reinforcement learning (RL), and R1-VL-2B, utilizing Step-wise Group Relative Policy Optimization (StepGRPO). The comparative analysis indicates clear trends consistent with our earlier findings in the InternVL series. As shown in Figure 6, Qwen2.5VL-7B exhibits superior Step Accuracy (56.63%) compared to MM Eureka-7B (52.39%). Similarly, Qwen2VL-2B outperforms R1-VL-2B in Path Validity Rate (42.87% versus 35.72%) and Path Coverage Score (61.63% versus 50.48%), demonstrating significant performance drops associated with post-training methods. Although certain metrics like Key Step Coverage show modest improvements in post-trained models (MM Eureka-7B: 36.66% vs. Qwen2.5VL-7B: 31.39%), the overall pattern emphasizes a general reduction in multimodal coherence and visual-centric reasoning effectiveness post-training. These findings align with observations from the InternVL models discussed in the main text and reinforce the conclusion that various post-training approaches, despite improving alignment to specific tasks, may impair generalization, particularly in visual-centric and dynamic reasoning tasks.

# B  MORE EXPLORATION

## B.1  HUMAN PERFORMANCE

To estimate human performance, we recruited four undergraduate students who had received systematic training in physics and were familiar with fundamental physical concepts. Each student was asked to solve the same 100 instances used in our closed-source model evaluation. Unlike other benchmarks, MVPBench is formulated as a visual question answering (VQA) task, and the evaluation of *quality* and *efficiency* relies on the generation of detailed, step-by-step reasoning chains. Therefore, our human performance assessment focuses solely on the *diversity* metric. For each instance, students were provided with the question, answer, image(s), and annotated key reasoning steps. They were instructed to produce as many distinct reasoning chains as possible that could lead to the correct answer by covering all the provided key steps. The resulting outputs were then used to compute the diversity scores.

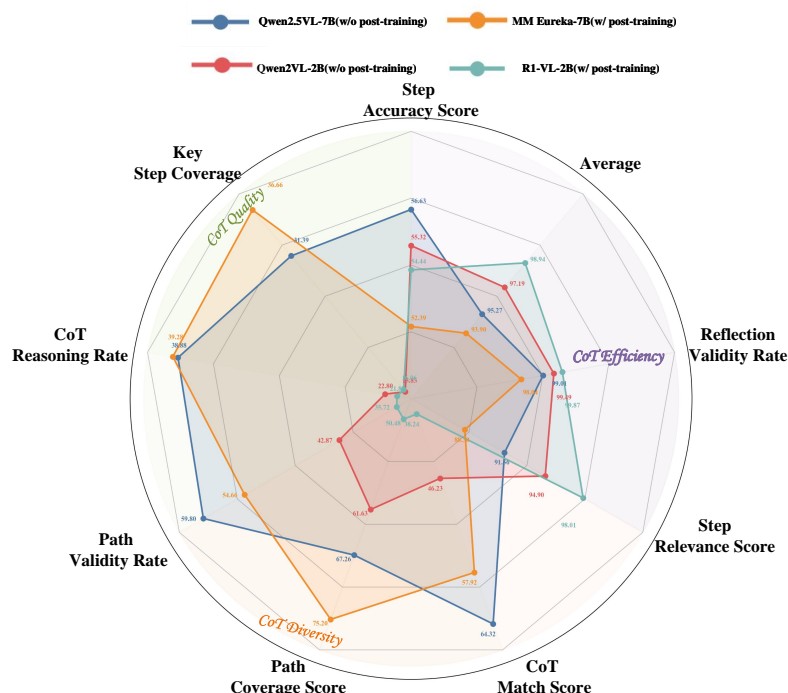

**Figure 6 CoT Performance of MLLMs with post-training versus without post-training.** Qwen2.5VL-7B and Qwen2VL-2B represent models without post-training, whereas MM Eureka-7B and R1-VL-2B denote their post-trained counterparts.Please note that each metric axis has its own independent scale.The results clearly indicate that post-training fails to enhance the reasoning performance of models and degrades it.

**Table 5 Expanded Subcategory-level Evaluation of CoT Reasoning in MVPBench: Closed-Source Models and Human Baselines.** We present a detailed subcategory-level evaluation of CoT reasoning along the dimensions of *Quality*, *Diversity*, and *Efficiency*, comparing closed-source MLLMs with human performance on MVPBench.

| Model | Phys-Experiment | | | Phys-Problems | | | Spatial-Relation | | | Dyn-Prediction | | |
|---|---|---|---|---|---|---|---|---|---|---|---|---|
| | Quality | Diversity | Efficiency | Quality | Diversity | Efficiency | Quality | Diversity | Efficiency | Quality | Diversity | Efficiency |
| *Human Performance* | | | | | | | | | | | | |
| | - | 98.72 | - | - | 96.42 | - | - | 99.13 | - | - | 95.76 | - |
| *Closed-source MLLMs* | | | | | | | | | | | | |
| Gemini-2.5-flash-preview-04-17 Deepmind (2024) | 61.85 | 68.64 | 100.00 | 63.37 | 39.10 | 85.56 | 28.36 | 73.04 | 93.26 | 46.62 | 48.00 | 100.00 |
| Grok3xAI (2025) | 43.85 | 65.54 | 87.60 | 58.16 | 72.26 | 78.50 | 50.60 | 58.72 | 85.66 | 58.16 | 58.59 | 99.33 |

## B.2 Error Analysis

To delve into the fine-grained predictions, we select the best-performing MLLM, GPT-4oOpenAI (2024), to understand its modes of success and failure. Our proposed CoT evaluation strategy has produced a detailed assessment of model output, including step-wise scores and explanation, reducing extensive manual effort in identifying and analyzing errors. As shown in Figure 7, we conduct our analysis on the two-step output from the CoT evaluation across the entire dataset, focusing on two key dimensions.

**Reasoning Errors Dominate Across Subcategories.** In particular, the proportion of visual perception errors in the physics-related subset is remarkably low—only 2.12% and 1.98% under single- and multi-image inputs, respectively. This finding contrasts with prior observations in MathVerse Zhang et al. (2024), highlighting the distinct characteristics of our benchmark. We posit that, within our dataset, GPT-4o is generally able to perceive the visual input correctly, but often fails during the reasoning process, leading to incorrect final answers.

**Spatial-Relation Emerge as a Major Source of Perception Failures.** In the spatial-relation subset, visual perception errors account for a striking 33.01% and 26.41% under single- and multi-image settings, respectively—substantially higher than in other subsets. This aligns with earlier findings that both closed-source and open-source MLLMs consistently perform worst on spatial relation tasks in terms of the quality metric. These results further support our initial hypothesis: current models struggle significantly with visual grounding when interpreting spatial relationships, underscoring a persistent bottleneck in multimodal understanding.

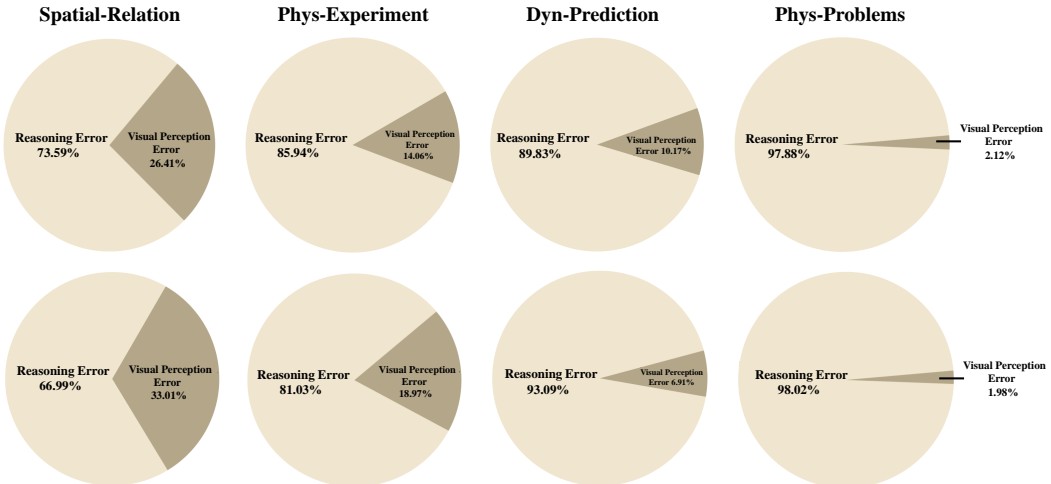

**Figure 7** **Distribution of GPT-4o OpenAI (2024) Errors across Different Types.** We report the error distribution of GPT-4o on MVPBench, categorized into two types: *Visual Perception Errors* and *Reasoning Errors*, across four representative subcategories. The first row illustrates the error distribution under single-image input settings, while the second row presents results under multi-image inputs.

## C  MORE DATASET DETAILS

### C.1  DATA COLLECTION

To support the evaluation of multimodal physical reasoning, we constructed a diverse and well-structured dataset spanning four distinct subdomains: (1) physics experiment videos, (2) conceptual physics questions, (3) spatial reasoning images, and (4) dynamic physical scene videos. **The annotation process was carried out between March 28 and May 14, 2025, by a team of 31 annotators with backgrounds in physics, science education, and computer vision.** Each data modality followed a carefully designed protocol to ensure quality, consistency, and relevance to downstream reasoning tasks.

**Table 6** Annotation summary across the four data modalities.

| Data Type | Sample Count | Average Length | Annotators |
|---|---|---|---|
| Physics Experiment Videos | 440 | 60 seconds | 16 |
| Conceptual Physics Problems | 320 | 200 words | 7 |
| Spatial Reasoning Images | 400 | 1 image | 4 |
| Dynamic Scene Videos | 100 | 2 seconds | 4 |

**Physics Experiment Videos.** This subset consists of 440 real-world videos sourced primarily from science education creators on Bilibili, such as "Lighthouse Laboratory" and "Interesting physics in life". These videos depict demonstrative physics experiments across domains including mechanics, optics, electromagnetism, and thermodynamics. Each video was segmented into a sequence of 3 to 5 keyframes capturing critical steps of a physical process. Annotators provided a natural language description for the initial state, intermediate key steps (each with conclusions), and a final outcome.

Visual markers (e.g., arrows, labeled objects) were optionally added to enhance clarity. Multiple plausible reasoning chains were manually curated to reflect different logical paths. All samples underwent double annotation with inter-annotator agreement checks and periodic expert reviews. The average duration per video was approximately 60 seconds.

**Conceptual Physics Problems.** This subset includes 320 multiple-choice and short-answer physics questions derived from high school curricula and online education platforms. Each item was manually adapted to include visual support (e.g., diagrams or plots), and transformed into a question-answer format with structured reasoning chains. Annotators selected questions where visual content was essential to reasoning, added visual cues to images (e.g., red dots, arrows), and reformulated options into logical deduction steps. Stepwise reasoning was expressed using Markdown-compatible mathematical expressions to support neural symbolic processing. The annotation reference document for this task was "MCoT-phytest.docx." All data underwent double annotation and review for logical soundness, visual accuracy, and completeness. On average, each problem included 200 words of reasoning and annotations.

**Spatial Reasoning Images.** This subset comprises 400 images curated from public domain resources such as Unsplash, Pixabay, and Archive.org. It addresses four categories of spatial reasoning: directional relations, distance estimation, perspective transformations, and topological connectivity. Annotators formulated tasks such as "What direction is object A facing?" or "From the first-view perspective of object A, where is object B?", using generic language to avoid lexical leakage. Key steps were illustrated using labeled visual cues and blue/red markings. Logical reasoning was written in natural language chains, each step tied to a specific visual cue or interpretation. Annotation was guided by the document "MCoT-spatial.docx" and performed by 4 annotators with experience in spatial cognition and vision tasks.

**Dynamic Physical Scene Videos.** The final subset includes 100 short video clips (average duration 2 seconds) selected from the PhysBench dataset. The tasks focus on predicting physical dynamics, such as object collision trajectories, liquid flow directions, and stability outcomes. Annotators extracted representative keyframes from each video and documented the physical evolution using a minimal chain of reasoning steps. For instance, a liquid falling through barriers would be annotated by highlighting key deflection events and predicting the final compartment of flow. Problems were written in standardized English using referential expressions (e.g., object A, path B). All dynamic samples followed the procedure detailed in "dynamic-prediction.docx," and were annotated by 4 individuals with expertise in physics simulation and time-series interpretation.

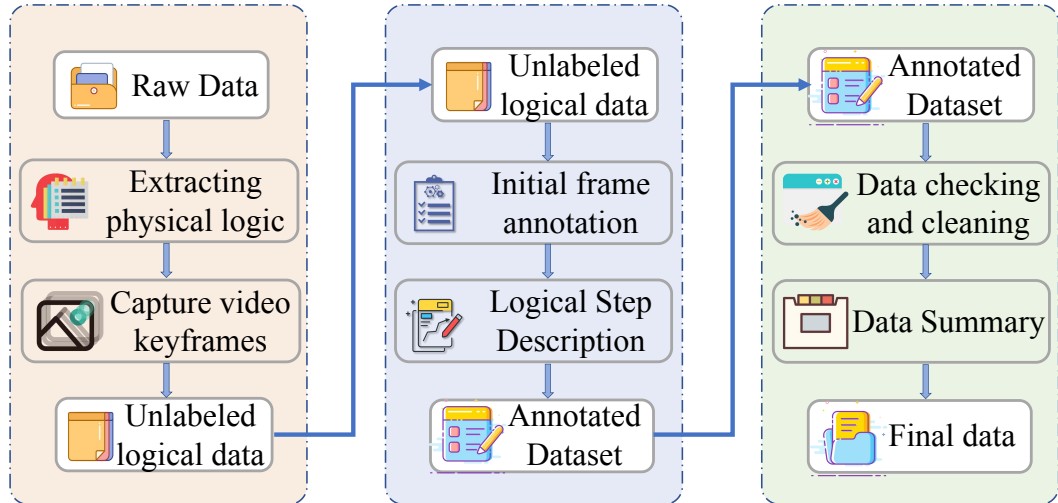

**Figure 8  Data collection process.** Initially, all visual and textual data undergo rigorous manual selection to ensure accuracy and relevance. Subsequently, expert annotators manually identify and highlight key objects and events, marking them visually with indicators such as arrows, and provide precise textual annotations for each critical step. Finally, multiple reasoning chains and key step annotations are meticulously constructed and validated manually, ensuring high-quality, reliable data for evaluating multimodal reasoning capabilities.

## C.2 DETAILED OF MVPBENCH COMPOSITION

**Physics Experiments.** The Physics Experiments subset of MVPBench contains a curated collection of 400 experimental questions, each designed to evaluate a model's understanding of sequential physical processes through multi-step visual inference. These experiments span five fundamental categories: Mechanics (222 questions), Thermodynamics (90 questions), Electromagnetism (42 questions), Optics (33 questions), and Kinematics (13 questions). Models must visually interpret the sequence of events and logically deduce the physical processes involved. In Mechanics tasks, models must interpret scenarios involving force interactions and motion, whereas Thermodynamics problems require reasoning about heat transfer and energy dynamics. Electromagnetism experiments involve interpreting visual representations of electric circuits and magnetic fields. Optics tasks test understanding of light behavior, reflection, and refraction, while Kinematics scenarios focus on analyzing motion trajectories and velocities. These tasks collectively ensure that the evaluated models develop comprehensive visual reasoning abilities similar to how humans mentally simulate physical experiments.

**Physics Problems.** The Physics Problems subset contains a total of 311 challenging, visually grounded physics questions, primarily sourced from academic examination databases such as Chinese Gaokao physics questions, the International Physics Olympiad (IPhO), and Chinese Mock Examinations at Various Levels, further augmented by additional questions from the PhysReason-mini dataset. These problems span five core physics categories: Mechanics (58 questions), Thermodynamics (56 questions), Electromagnetism (90 questions), Optics (53 questions), and Kinematics (54 questions). Mechanics questions may involve complex analysis of force interactions or equilibrium scenarios, while Thermodynamics problems often present visual cues related to heat exchange and energy conversion processes. Electromagnetism tasks require reasoning about visually depicted electric circuits and magnetic field interactions. Optics questions focus on image formation, lens behavior, and optical phenomena, and Kinematics challenges typically demand interpretation of visual trajectories, acceleration, and velocity vectors. This detailed structuring and multimodal approach aim to assess models' capabilities in accurately interpreting visual information and applying advanced reasoning to solve intricate physics problems.

**Spatial Relations.** The Spatial Relations subset assesses spatial perception through 400 carefully designed questions, divided into four specific subcategories. (1) Direction Judgment (100 questions): This subcategory requires models to accurately determine the relative directional positioning of various objects within a scene, emphasizing an understanding of spatial orientation and relational positioning. (2) Distance Estimation (100 questions): Tasks here involve estimating the distance and depth relations between objects or between objects and the camera viewpoint, highlighting the importance of accurate depth perception and visual estimation skills. (3) First-view Transformation (100 questions): This subcategory challenges models to reason about spatial directions from an egocentric viewpoint, simulating real-world scenarios where orientation judgments are made from a first-person perspective. (4) Topological Relation Judgment (100 questions): This category focuses specifically on assessing the reachability and connectivity within directed graphs, using images constructed through graphical editing tools. Overall, this subset is designed to rigorously evaluate models' capabilities in processing complex spatial scenarios and performing accurate spatial reasoning, reflecting essential cognitive processes used in navigating and interpreting real-world visual environments.

**Dynamic Prediction.** The Dynamic Prediction subset comprises 100 tasks designed to evaluate the predictive capabilities of models regarding dynamically evolving physical interactions, structured into four subcategories: (1) Multi-object Collision (25 questions): This category requires models to predict outcomes involving interactions among multiple objects, such as collisions, considering momentum, energy transfer, and motion trajectories. (2) Liquid Diversion (25 questions): Tasks involve predicting fluid paths through variously configured channels or obstacles, necessitating models to understand fluid dynamics visually. (3) Physical State Prediction (25 questions): These problems challenge models to anticipate changes in the physical states of objects, such as transitions between solid, liquid, and gas phases, based on visual cues and temporal sequences. (4) Shadow Transformation Prediction (25 questions): This subcategory assesses the ability of models to predict and interpret the changes in shadows cast by objects due to movements or shifts in light sources, requiring sophisticated temporal and spatial reasoning. These tasks collectively aim to test models' capacity to interpret and forecast dynamic physical phenomena, thereby closely replicating human cognitive processes involved in visual prediction and temporal reasoning.

## C.3 DATA ANALYSIS

Table 7 presents core statistics of the MVPBench dataset, which consists of 1,211 samples with a total of 4,701 images, covering both unique and repeated images. Each question and corresponding answer is distinct, underscoring the dataset's broad range and depth across various physical reasoning scenarios. Furthermore, question lengths display considerable variation, with some reaching up to 100 words, though the majority of questions are moderately sized. Answers generally involve multiple reasoning steps, reflecting a significant complexity level within the dataset. Notably, the dataset includes multiple Image-CoTs per sample—visual chains of thought specifically crafted as input to guide and assess model reasoning processes. The average number of Image-CoTs per sample is approximately 3.90, with some samples containing up to 5, ensuring rich visual context for enhanced multimodal reasoning. Additionally, each sample captures several chains of thought, facilitating the evaluation of multi-path reasoning capabilities.

The dataset includes multiple subsets(Figure 9), with Physics Experiments and Spatial Relations forming the most significant components, emphasizing sequential reasoning through multi-step physical processes and complex spatial perception tasks, respectively. Additionally, a substantial contribution from the Physics Problems subset highlights the emphasis on advanced textual comprehension in our benchmark. The inclusion of Dynamic Prediction subset further ensures comprehensive evaluation under conditions involving temporal changes and challenging visual contexts. Collectively, the structured distribution across these subsets fosters a balanced assessment of diverse visual reasoning capabilities crucial for a robust understanding of physical phenomena.

| Statistic | Value |
|---|---|
| Total samples | 1,211 |
| Total images | 4,701 |
| Unique images | 4,688 |
| Unique questions | 1,211 |
| Unique answers | 1,211 |
| Max. question length | 100 |
| Avg. question length | 28.01 |
| Max. answer steps | 9 |
| Avg. answer steps | 2.93 |
| Max. Image-CoTs per sample | 5 |
| Avg. Image-CoTs per sample | 3.90 |
| Max. reasoning paths | 16 |
| Avg. reasoning paths | 2.67 |

**Table 7  Key statistics of MVPBench.** Summarizes dataset size, question/answer properties, and multi-path reasoning annotations for evaluating complex reasoning in MLLMs.

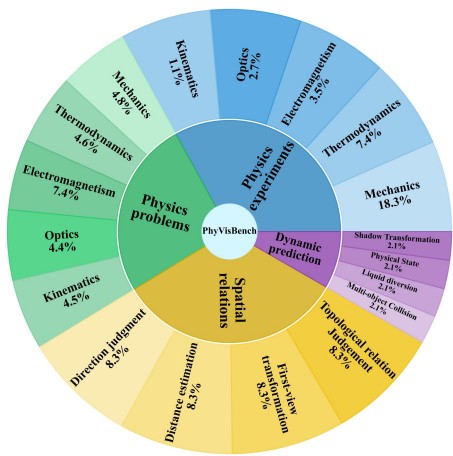

**Figure 9  Category distribution in MVPBench.** Covers 4 major reasoning categories and 18 fine-grained subcategories.

## C.4 ADDITIONAL STATISTICS OF DATASET

This section presents further statistical analyses to offer deeper insights into the composition and characteristics of the dataset. As Shown in Figure 10, Figure (a) provides an overview of the distribution of physics concepts encountered within the reasoning steps. It reveals that certain foundational concepts such as "light," "force," and "pressure" are notably prevalent, indicating their central importance within the reasoning processes of datasets. The distribution of these concepts emphasizes their relative significance and highlights the necessity for models to grasp core physics principles robustly. Figure (b) illustrates the distribution of query word counts through a histogram accompanied by a kernel density estimation curve, effectively capturing the general complexity and length patterns of the queries. The data suggests a predominance of moderately sized questions, though there exists a notable tail extending towards longer, more complex queries, underscoring the variety in question complexity.

The distribution of reasoning chains, depicted in Figure (c), offers valuable insights into the diversity of dataset in reasoning paths per sample. Most samples incorporate one or two distinct chains, highlighting the presence of alternative reasoning pathways. Nonetheless, there is a non-negligible proportion of instances with several reasoning chains, indicating complexity and diversity in the reasoning processes required by the dataset. Figure (d) examines the distribution of reasoning steps per sample. The analysis indicates variability in the complexity of the reasoning tasks, with most samples containing a moderate number of steps. This reflects the balance of dataset between simplicity and complexity, essential for comprehensively evaluating reasoning proficiency.

Reasoning complexity, as shown in Figure (e), combines reasoning steps and the number of reasoning chains to provide a composite indicator of overall reasoning demand. The distribution confirms that while many instances involve relatively straightforward reasoning, a meaningful subset presents significant complexity, requiring intricate, multi-faceted reasoning capabilities. Finally, Figure (f) explores the distribution of images included per sample. It demonstrates a balanced use of visual information, with most samples featuring several images to guide visual reasoning tasks effectively. This emphasis on visual context underscores the intent to robustly assess models' capabilities in interpreting and reasoning about visually grounded information.

We further compute the ratio between the Relevant Steps (Generated) and the Key Steps (Ground Truth) to examine the step-level differences between the annotated reasoning chains and those generated by the models. The results are summarized in table 8.

**Table 8** Comparison of Relevant and Key Steps in Reasoning Chains.

| Model | Ratio |
|---|---|
| *Open-source MLLMs* | |
| LLaVA-OV-72B | 3.87 |
| LLaVA-CoT | 4.29 |
| InternVL2.5-78B | 5.15 |
| InternVL2.5-78B-MPO | 5.18 |
| InternVL3-78B | 5.01 |
| InternVL3-78B-Instruct | 5.02 |
| Qwen2.5-VL-7B | 5.14 |
| Qwen2.5-VL-72B | 5.03 |
| QVQ-72B | 6.75 |
| *Closed-source MLLMs* | |
| GPT-4o | 5.46 |
| OpenAI o3 | 4.93 |
| Claude 3.7 Sonnet | 6.09 |

C.5    ANALYSIS EGARDING THE EVALUATION COST

We acknowledge that the proposed multi-path visual reasoning evaluation framework may incur additional token consumption and time overhead in practical applications. To address these concerns, we have conducted a comprehensive and detailed statistical analysis of the evaluation cost.

Specifically, we summarized and analyzed all API calls on the complete MVPBench dataset using the official GPT-4o pricing ( input 2.50$ / 1M tokens, output 10.00$ / 1M tokens). The detailed results are summarized in the table 9.

The statistics above indicate that although our evaluation method introduces more sophisticated assessment dimensions, the overall economic cost remains within a reasonable range. The total cost for evaluating all 1211 samples is approximately $29.

With an average cost per sample of approximately $0.0060, the evaluation cost per 1000 samples is about $6, demonstrating that the evaluation expenses are manageable and affordable. This makes our evaluation method economically feasible even for large-scale testing scenarios. Additionally, each sample evaluation takes on average only 5.28 seconds, resulting in a total assessment time of merely

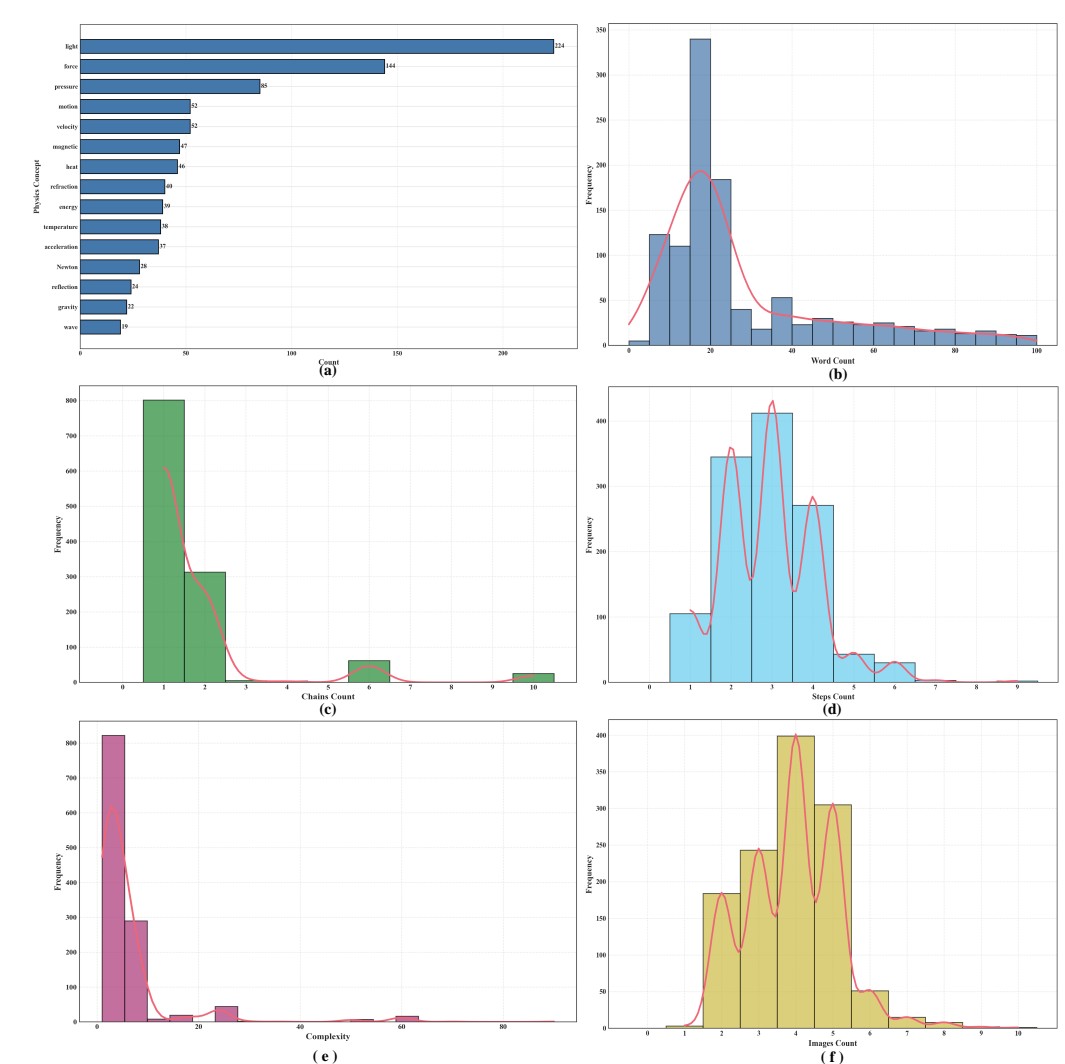

**Figure 10** **Additional statistic.** Figure a is the Physics Concepts Distribution, this horizontal bar chart shows the frequency of physical concepts that appear in the reasoning steps. The Y-axis represents physical concepts , and the X-axis represents the number of occurrences. Figure b is the Query Word Count Distribution, this histogram shows the distribution of the number of words in the questions. The X-axis represents the number of words, and the Y-axis represents the frequency. Figure c is the Reasoning Chains Distribution, this histogram shows how many different reasoning paths each sample contains. Figure d is the Reasoning Steps Distribution, this histogram shows how many reasoning steps each sample contains. The X-axis represents the number of steps, and the Y-axis represents the frequency. Figure e is the Reasoning Complexity Distribution, this histogram shows the distribution of complexity indicators. Complexity is defined as the number of reasoning steps × the number of different reasoning paths. Figure f is the Sample Images Distribution, this histogram shows how many images each sample contains.

7.1 hours on one cheap GPU(Even parallel acceleration can be achieved through multi-terminal operation) for the entire benchmark. PhysReasonZhang et al. (2025c), by contrast,inspect ≈ 8.1 annotated steps (≈ 441 answer tokens) per problem, invoking the scorer for each and driving the per-item budget to ≈ 1.6k tokens—about \$0.048, eight times MVPBench—so its authors released a trimmed 200-question mini set to keep costs in check. MME-CoTJiang et al. (2025a) is similarly token-hungry: its three-axis scheme slices the chain-of-thought, adds a reflection sweep, and repeats for robustness, greatly increasing latency. MVPBench attains the same analytical breadth—CoT Quality, Efficiency, and a dedicated CoT Diversity metrics—yet requires only one forward pass. Quality and Efficiency are judged directly from the full answer, while Diversity leverages a compact directed-graph encoding of all admissible reasoning paths, a single alignment thus yields fine-grained

**Table 9** Evaluation cost and efficiency.

| Metric | Num |
|---|---|
| Average Prompt tokens per sample | 984.47 tokens |
| Average Completion tokens per sample | 354.31 tokens |
| Average Tokens per sample | 1338.78 tokens |
| Average Time Consumption per sample | 5.28 s |
| Average Cost per call | $0.006004 |

coverage without iterative decomposition. The result is richer diagnostics at roughly one-tenth the cost of PhysReason and markedly less time than MME-CoT. Although our method increases the use of tokens, it brings significant benefits, proving that the additional token usage and time consumption are justified.

In summary, our evaluation method achieves a favorable balance between additional cost and the substantial scientific benefits provided. We again thank the reviewers for raising this important concern.

## D MORE QUALITATIVE EXAMPLES

In this section, we present additional qualitative examples to offer deeper insights into our evaluation framework and illustrate the practical application of our metrics. The examples are organized into three subsections, each highlighting a specific evaluation dimension: Quality, Diversity, and Efficiency.

### D.1 EXAMPLES OF QUALITY EVALUATION.

This subsection includes eight illustrative examples, as shown in Figure 11 to Figure 18, focusing specifically on evaluating the quality of the reasoning provided by various models. Each example showcases detailed analyses of model responses, assessing Step Accuracy Score (SAS) and Key Step Coverage (KSC).

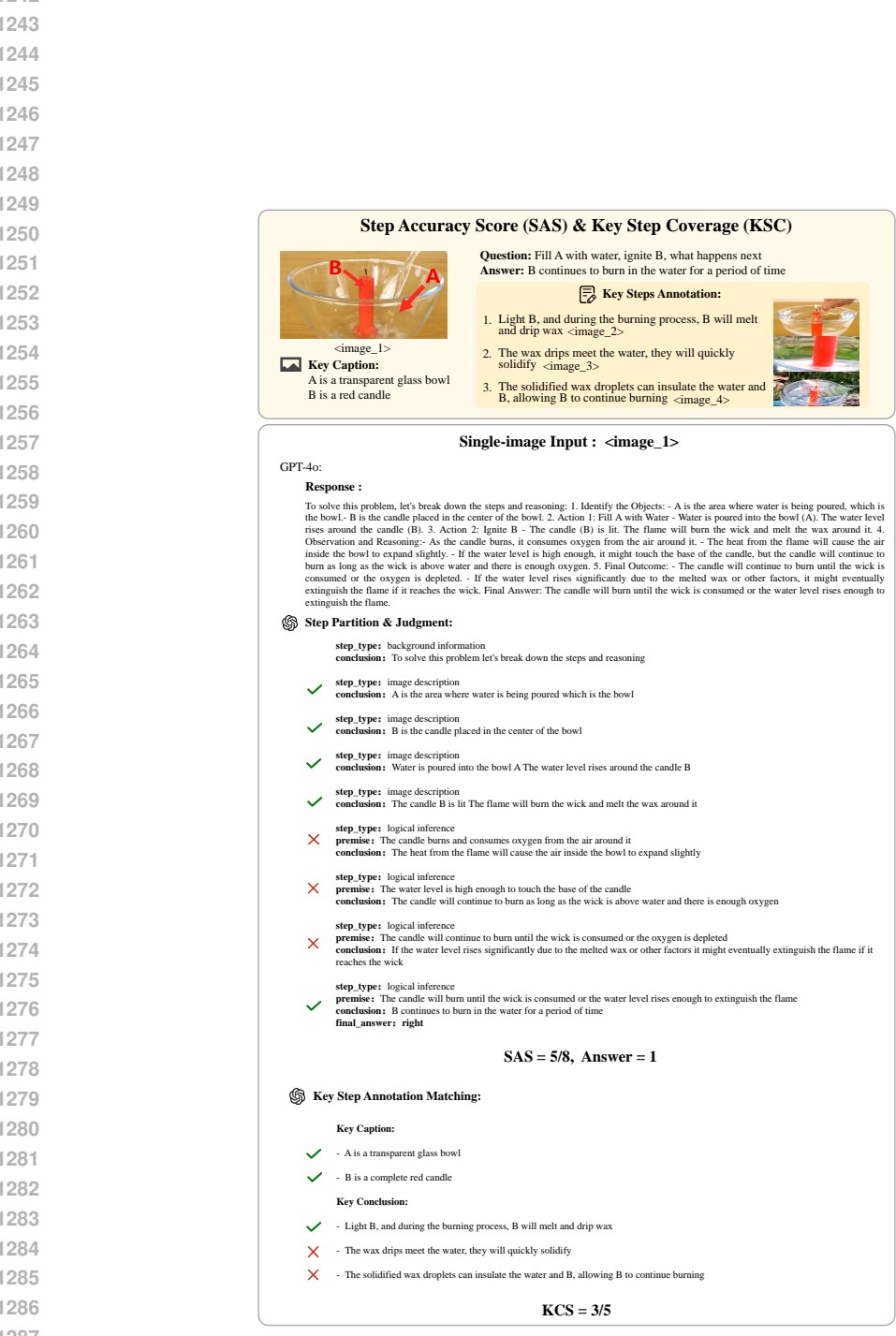

**Figure 11  Examples of Single-Image SAS–KSC Evaluation on Physics Experiment Data.**

**Multi-image Input : <image_1,2,3,4>**

GPT-4o:

**Response :**

To analyze what happens step by step, let's break down the process based on the images and the instructions: Step 1: Fill A with water - Observation: A is a transparent bowl, and water is being poured into it. The red candle (B) is already placed upright in the center of the bowl. - Reasoning: The water level rises in the bowl, but the candle remains above the water surface because it is designed to float or is fixed in place. Step 2: Ignite B - Observation: The wick of the red candle (B) is lit, producing a flame. - Reasoning: The candle burns as usual, with the flame consuming the wax and wick. The flame is unaffected by the surrounding water because the wick and wax are above the waterline. Step 3: What happens next? - Observation: The candle continues to burn, and the heat from the flame melts the wax near the wick. The melted wax may drip down or pool around the base of the candle. -Reasoning: The water in the bowl acts as a heat sink, preventing the candle from overheating or melting excessively. The flame remains stable as long as the wick is above the water and has access to oxygen. Step 4: Final Outcome - Observation: The candle burns steadily while floating or standing in the water. The water prevents the melted wax from spreading too far, and the flame remains unaffected. - Reasoning: The setup creates a visually interesting effect where the candle appears to burn "on water." The water also adds safety by reducing the risk of fire spreading. Final Answer: When you fill A (the bowl) with water and ignite B (the candle), the candle burns steadily while floating or standing in the water. The water acts as a heat sink and safety barrier, creating an intriguing visual effect of a burning candle surrounded by water.

**Step Partition & Judgment:**

> **step_type:** background information
> **conclusion:** Fill A with water ignite B what happens next

> **step_type:** background information
> **conclusion:** Object A is a transparent glass bowl and Object B is a complete red candle

✓ **step_type:** image description
> **conclusion:** A is a transparent bowl and water is being poured into it The red candle B is already placed upright in the center of the bowl

✓ **step_type:** logical inference
> **premise:** The water level rises in the bowl but the candle remains above the water surface because it is designed to float or is fixed in place
> **conclusion:** The candle remains above the water surface

✓ **step_type:** image description
> **conclusion:** The wick of the red candle B is lit producing a flame

✓ **step_type:** logical inference
> **premise:** The candle burns as usual with the flame consuming the wax and wick The flame is unaffected by the surrounding water because the wick and wax are above the waterline
> **conclusion:** The flame is unaffected by the surrounding water

✓ **step_type:** image description
> **conclusion:** The candle continues to burn and the heat from the flame melts the wax near the wick The melted wax may drip down or pool around the base of the candle

✗ **step_type:** logical inference
> **premise:** The water in the bowl acts as a heat sink preventing the candle from overheating or melting excessively The flame remains stable as long as the wick is above the water and has access to oxygen
> **conclusion:** The flame remains stable as long as the wick is above the water

✓ **step_type:** image description
> **conclusion:** The candle burns steadily while floating or standing in the water The water prevents the melted wax from spreading too far and the flame remains unaffected

✓ **step_type:** logical inference
> **premise:** The setup creates a visually interesting effect where the candle appears to burn on water The water also adds safety by reducing the risk of fire spreading
> **conclusion:** The candle appears to burn on water

✓ **step_type:** logical inference
> **premise:** When you fill A the bowl with water and ignite B the candle the candle burns steadily while floating or standing in the water. The water acts as a heat sink and safety barrier creating an intriguing visual effect of a burning candle surrounded by water
> **conclusion:** B continues to burn in the water for a period of time
> **final_answer:** right

**SAS = 8/9, Answer = 1**

**Key Step Annotation Matching:**

**Key Caption:**

✓ - A is a transparent glass bowl

✓ - B is a complete red candle

**Key Conclusion:**

✓ - Light B, and during the burning process, B will melt and drip wax

✗ - The wax drips meet the water, they will quickly solidify

✓ - The solidified wax droplets can insulate the water and B, allowing B to continue burning

**KCS = 4/5**

**Figure 12** Examples of Multi-Image SAS–KSC Evaluation on Physics Experiment Data: Enhanced Visual Context Enables Richer Reasoning and Performance Gains.

none

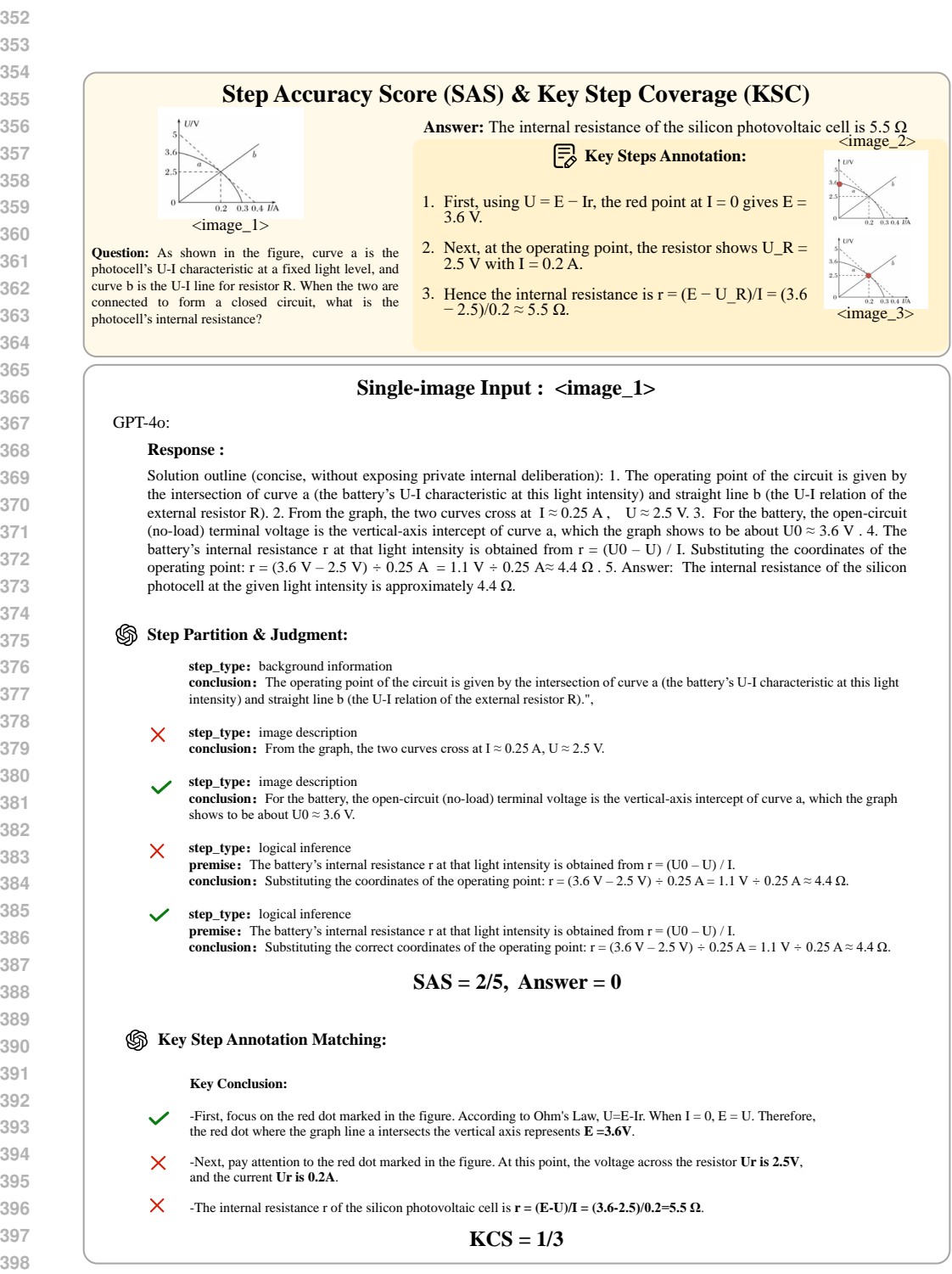

**Figure 13 Examples of Single-Image SAS–KSC Evaluation on Physics Problems Data with GPT-4o.**

**Multi-image Input : <image_1,2,3>**

GPT-4o:

**Response :**

When the silicon photocell is connected to the resistor R, the working point of the circuit is the intersection P of curve a (photocell U-I characteristic) and line b (resistor U-I characteristic). 1. From the graph this intersection is at current I ≈ 0.20 A terminal voltage U ≈ 2.5 V. 2. The open-circuit (no-load) voltage of the photocell, i.e. the intercept of curve a on the U-axis, is about E ≈ 3.6 V. 3. For a source with emf E, terminal voltage U and current I, the internal resistance r = (E − U) / I. 4. Substituting the values read from the graph , r = (3.6 V − 2.5 V) / 0.20 A ≈5.5Ω. 5. Therefore, the internal resistance of the silicon photocell at this light intensity is about r ≈ 5.5 Ω.

**Step Partition & Judgment:**

**step_type：** background information
**conclusion：** When the silicon photocell is connected to the resistor R, the working point of the circuit is the intersection P of curve a (photocell U-I characteristic) and line b (resistor U-I characteristic).

✓ **step_type：** image description
**conclusion：** From the graph this intersection is at current I ≈ 0.20 A and terminal voltage U ≈ 2.5 V.

✓ **step_type：** image description
**conclusion：** The open-circuit (no-load) voltage of the photocell, i.e. the intercept of curve a on the U-axis, is about E ≈ 3.6 V.

✓ step_type: background information
conclusion: For a source with emf E, terminal voltage U and current I, the internal resistance r is r = (E − U) / I.

✓ **step_type：** logical inference
**premise：** E = 3.6 V, U = 2.5 V, I = 0.20 A
**conclusion：** Substituting the values read from the graph, r = (3.6 V − 2.5 V) / 0.20 A, r ≈ 5.5 Ω.

✓ **step_type：** logical inference
**premise：** E = 3.6 V, U = 2.5 V, I = 0.20 A.
**conclusion：** Therefore, the internal resistance of the silicon photocell at this light intensity is about r ≈ 5.5 Ω.

### SAS = 6/6,  Answer =1

**Key Step Annotation Matching:**

**Key Conclusion:**

✓ -First, focus on the red dot marked in the figure. According to Ohm's Law, U=E-Ir. When I = 0, E = U. Therefore, the red dot where the graph line a intersects the vertical axis represents **E =3.6V**.

✓ -Next, pay attention to the red dot marked in the figure. At this point, the voltage across the resistor **Ur is 2.5V**, and the current **Ur is 0.2A**.

✓ -The internal resistance r of the silicon photovoltaic cell is **r = (E-U)/I = (3.6-2.5)/0.2=5.5 Ω**.

### KCS = 3/3

**Figure 14** Examples of Multi-Image SAS–KSC Evaluation on Physics Tasks with GPT-4o: Enhanced Visual Context Boosts Reasoning and Performance.

**Multi-image Input : <image_1,2,3>**

Internvl3-78B:

**Response :**

To determine the internal resistance of the silicon photocell when it forms a closed loop with the resistor R at a certain light intensity, we need to use the given U-I graph. 1. Identify the Open-Circuit Voltage $U_{oc}$ : The open-circuit voltage is the voltage across the battery when no current flows (i. e., $I = 0$ ).From the graph, when $I = 0$, $U = 3.6$ V. Therefore, $U_{oc}$ = 3.6 V. 2. Determine the Short-Circuit Current $I_{sc}$ :The short-circuit current is the current when the voltage across the battery is zero (i. e., $U = 0$). From the graph, when $U = 0$, $I = 0.4$ A. Therefore, $I_{sc}$ = 0.4 A. 3. Calculate the Internal Resistance $r$ :The internal resistance of the battery can be calculated using the formula: $r = \frac{U_{oc}}{I_{sc}}$. Substituting the values: $r = \frac{3.6 \text{ V}}{0.4 \text{ A}} = 9$ Ω$. 4. The internal resistance of the silicon photocell at this light intensity is 9 Ω.

**Step Partition & Judgment:**

✔ **step_type：** image description
**conclusion：** The open-circuit voltage is the voltage across the battery when no current flows (i.e., I = 0). From the graph, when I = 0, U = 3.6 V.

✘ **step_type：** image description
**conclusion：** The short-circuit current is the current when the voltage across the battery is zero (i.e., U = 0). From the graph, when U = 0, I = 0.4 A.

✘ **step_type：** logical inference
**premise：** U_oc = 3.6 V. I_sc = 0.4 A.
**conclusion：** The internal resistance of the battery can be calculated using the formula: r = U_oc / I_sc. Substituting the values: r = 3.6 V / 0.4 A = 9 Ω.

✘ **step_type：** logical inference
**premise：** U_oc = 3.6 V. I_sc = 0.4 A.
**conclusion：** The internal resistance of the silicon photocell at this light intensity is 9 Ω.

**SAS = 1/4,  Answer =0**

**Key Step Annotation Matching:**

**Key Conclusion:**

✔ -First, focus on the red dot marked in the figure. According to Ohm's Law, U=E-Ir. When I = 0, E = U. Therefore, the red dot where the graph line a intersects the vertical axis represents **E =3.6V**.

✘ -Next, pay attention to the red dot marked in the figure. At this point, the voltage across the resistor **Ur is 2.5V**, and the current **Ur is 0.2A**.

✘ -The internal resistance r of the silicon photovoltaic cell is **r = (E-U)/I = (3.6-2.5)/0.2=5.5 Ω**.

**KCS = 1/3**

**Figure 15** **Examples of Multi-Image SAS–KSC Evaluation on Physics Problems Data with InternVL2.5.**

## Step Accuracy Score (SAS) & Key Step Coverage (KSC)

**Question:** From A's perspective, which direction is B

**Answer:** B is located to the left rear of A

📝 **Key Steps Annotation:**

1. A is a yellow and black car at the top of the intersection
   B is a yellow and black car near the bottom  <image_2>

2. A faces the upper left, back toward the camera  <image_3>

3. B is located below, slightly to the right  <image_4>

4. From A's view, B is in the left rear  <image_5>

<image_1>

### Single-image Input : <image_1>

GPT-4o:

**Response :**

To determine the direction of object B from the first-person perspective of object A, follow these steps: 1.Identify the Orientation of Object A: - Object A is a vehicle facing towards the top left of the image, moving diagonally across the intersection. 2.Determine Object A's Forward Direction: - From the perspective of object A, the forward direction is towards the top left corner of the image. 3.Locate Object B Relative to Object A:- Object B is positioned towards the bottom of the image, moving upwards. 4. Determine the Direction from A to B: - From object A's perspective, object B is located to the right and slightly behind. 5.Final Direction: - Therefore, from the first-person perspective of object A, object B is to the right.

🔮 **Step Partition & Judgment:**

**step_type:** background information
**conclusion:** To determine the direction of object B from the first-person perspective of object A, follow these steps

✅ **step_type:** image description
**conclusion:** Object A is a vehicle facing towards the top left of the image, moving diagonally across the intersection

✅ **step_type:** logical inference
**premise:** Object A is a vehicle facing towards the top left of the image.
**conclusion:** From the perspective of object A, the forward direction is towards the top left corner of the image

❌ **step_type:** image description
**conclusion:** Object B is positioned towards the bottom of the image, moving upwards

❌ **step_type:** logical inference
**premise:** From the perspective of object A, the forward direction is towards the top left corner of the image. Object B is positioned towards the bottom of the image
**conclusion:** From object A's perspective, object B is located to the right and slightly behind

❌ **step_type:** logical inference
**premise:** From object A's perspective, object B is located to the right and slightly behind
**conclusion:** Therefore, from the first-person perspective of object A, object B is to the right

### SAS = 2/6,  Answer = 0

🔮 **Key Step Annotation Matching:**

**Key Conclusion:**

✅ - Identify object A and object B. Object A is a yellow and black car at the top of the intersection. Object B is a yellow and black car near the bottom of the intersection

❌ - Determine the direction of object A. Object A is located on the road with its back facing the camera, facing towards the upper left corner of the image

❌ - Determine the relative position of object B. Object B is located below Object A, slightly to the right

❌ - Determine direction. From the first person perspective of object A, object B is located in the left rear

### KCS = 1/4

**Figure 16** Examples of Single-Image SAS–KSC Evaluation on Spatial Relations Data with GPT-4o.

**Multi-image Input : <image_1,2,3,4,5>**

GPT-4o:

**Response :**

To determine the direction of object B from the perspective of object A, follow these steps: 1. Identify the Position of Object A: - Object A is the taxi at the top left of the image. 2. Draw a Line from Object A to Object B: Imagine or draw a straight line from the front of Object A (the taxi) to Object B (the other taxi at the bottom right). 3. Determine the Relative Direction: - From the perspective of Object A, determine which way Object B is located relative to the direction Object A is facing. Object B is positioned slightly to the right and directly in front of Object A. 4. Use the Blue Arrow for Clarification: - The blue arrow in the final image shows the direction from Object A to Object B, pointing forward and slightly to the right. 5. Conclusion: Object B is in front of and to the right of Object A. 6. Final Direction: From the perspective of Object A, Object B is in front and to the right.

**Step Partition & Judgment:**

    **step_type:** image description
    **conclusion:** Object A is the taxi at the top left of the image.

✓  **step_type:** image description
    **conclusion:** Object B is the taxi at the bottom right of the image.

✓  **step_type:** logical inference
    **premise:** Object A is the taxi at the top left of the image. Object B is the taxi at the bottom right of the image.
    **conclusion:** Draw a line from the front of Object A to Object B.

✗  **step_type:** logical inference
    **premise:** Draw a line from the front of Object A to Object B.
    **conclusion:** Object B is positioned slightly to the right and directly in front of Object A.

✓  **step_type:** image description
    **conclusion:** The blue arrow in the image shows the direction from Object A to Object B

✗  **step_type:** logical inference
    **premise:** The blue arrow in the image shows the direction from Object A to Object B.
    **conclusion:** The blue arrow points forward and slightly to the right.

✗  **step_type:** logical inference
    **premise:** Object B is positioned slightly to the right and directly in front of Object A.
    **conclusion:** From the perspective of Object A, Object B is in front and to the right.

**SAS = 3/7,  Answer = 0**

**Key Step Annotation Matching:**

    **Key Conclusion:**

✓  -Identify object A and object B. Object A is a yellow and black car at the top of the intersection. Object B is a yellow and black car near the bottom of the intersection.

✗  -Determine the direction of object A. Object A is located on the road with its back facing the camera, facing towards the upper left corner of the image.

✗  -Determine the relative position of object B. Object B is located below Object A, slightly to the right.

✗  -Determine direction. From the first person perspective of object A, object B is located in the left rear.

**KCS = 1/4**

**Figure 17** **Examples of Multi-Image SAS–KSC Evaluation on Spatial Relations Data with GPT-4o: Enhenced Visual Context Enables Richer Reasoning and Performance Gains.**

**Multi-image Input : <image_1,2,3,4,5>**

InternVL2.5:

**Response :**

To determine the direction of object B from the first-person perspective of object A, we need to analyze the images provided step-by-step: 1. Identify Object A's Perspective: - Object A is a vehicle located at the top left of the intersection, facing towards the right side of the image. The arrow pointing from object A indicates its forward direction. 2. Identify Object B's Position: Object B is another vehicle located at the bottom center of the intersection, facing upwards in the image. The arrow pointing from object B indicates its forward direction. 3. Determine Object A's Forward Direction:- From the first-person perspective of object A, the vehicle is facing towards the right side of the image, which corresponds to the east direction in a typical map orientation. 4. Determine Object B's Relative Position: - Object B is positioned below object A in the image. From object A's perspective, this means object B is located towards the south. 5. Analyze the Relative Direction: Since object A is facing east, and object B is positioned towards the south from object A's perspective, object B is located to the right and slightly behind object A. 6. Final Direction: From object A's perspective, object B is in the southeast direction.

**Step Partition & Judgment:**

    **step_type：** background information
    **conclusion：** To determine the direction of object B from the first-person perspective of object A, we need to analyze the images provided step-by-step

✔    **step_type：** image description
    **conclusion：** Object A is a vehicle located at the top left of the intersection, facing towards the right side of the image. The arrow pointing from object A indicates its forward direction.

✔    **step_type：** image description
    **conclusion：** Object B is another vehicle located at the bottom center of the intersection, facing upwards in the image. The arrow pointing from object B indicates its forward direction.

✘    **step_type：** logical inference
    **premise:** From the first-person perspective of object A, the vehicle is facing towards the right side of the image.
    **conclusion：** Object A is facing towards the east direction in a typical map orientation.

✘    **step_type：** logical inference
    **premise:** Object B is positioned below object A in the image.
    **conclusion：** From object A's perspective, this means object B is located towards the south.

✘    **step_type：** logical inference
    **premise：** Object A is facing east, and object B is positioned towards the south from object A's perspective.
    **conclusion：** Object B is located to the right and slightly behind object A.

✘    **step_type：** logical inference
    **premise：** From object A's perspective, object B is positioned towards the south
    **conclusion：** Object B is in the southeast direction.

**SAS = 2/7,  Answer = 0**

**Key Step Annotation Matching:**

    **Key Conclusion:**

✔    -Identify object A and object B. Object A is a yellow and black car at the top of the intersection. Object B is a yellow and black car near the bottom of the intersection.

✘    -Determine the direction of object A. Object A is located on the road with its back facing the camera, facing towards the upper left corner of the image.

✘    -Determine the relative position of object B. Object B is located below Object A, slightly to the right.

✘    -Determine direction. From the first person perspective of object A, object B is located in the left rear.

**KCS = 1/4**

**Figure 18**  **Examples of Multi-Image SAS–KSC Evaluation on Spatial Relations Data with InternVL3.**

### D.2 EXAMPLES OF DIVERSITY EVALUATION.

Figure 19 provides an example illustrating our diversity evaluation metrics - Path Validity Rate (PVR) and Path Coverage Score (PCS). We find that models vary in their ability to explore diverse reasoning paths when processing both single-image and multi-image inputs.

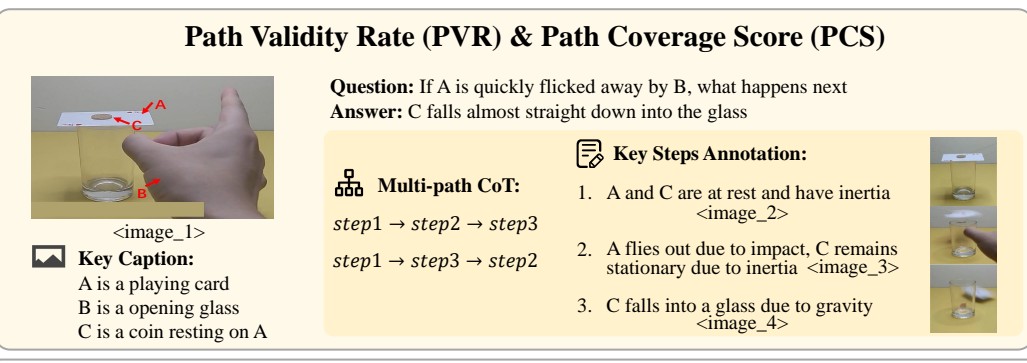

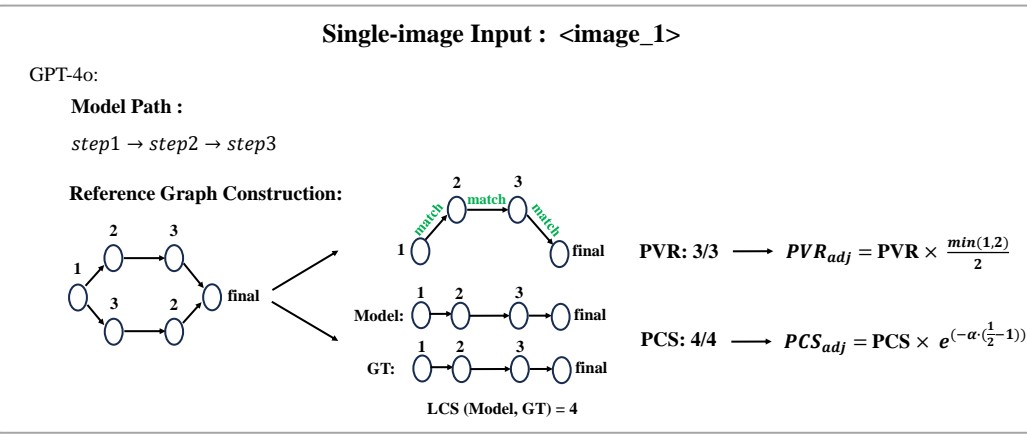

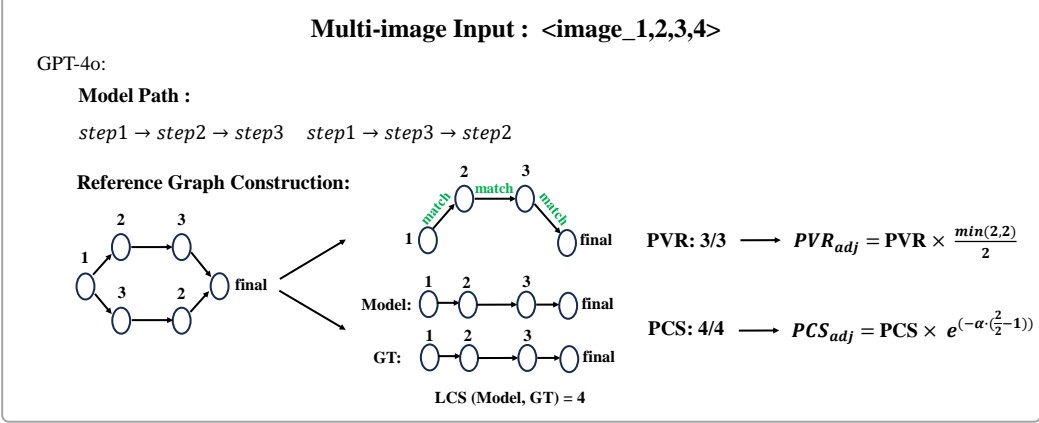

**Figure 19** **Examples of Single- and Multi-Image PVR-PCS Evaluation on Spatial Relations Data with GPT-4o.**

### D.3 EXAMPLES OF EFFICIENCY EVALUATION.

Figure 20 focuses on evaluating step relevance, clearly indicating how accurately and succinctly the models identify and utilize pertinent information from visual and textual inputs. Figure 21 specifically illustrates a scenario examining reflection validity, detailing the ability of model to avoid redundant or incorrect reasoning paths.

## Step Relevance Score (SRS)

<image_1>

**Key Caption:**
A is the black car on the right
B is the white car on the left

**Question:** From object A's view, which direction is object B
**Answer:** Object B is located to the left rear of Object A

📝 **Key Steps Annotation:**

1. A faces the camera from the side and is located in the middle of the road facing to the right <image_2>

2. B is located in the upper left corner of A, on the right side of the road <image_3>

3. From object A's view, B is located in the left rear direction <image_4>

### Single-image Input : <image_1>

Qwen2.5-VL-72B:

**Response :**

To determine the direction of object B from the first-person perspective of object A, let's follow these steps: 1. Identify Object A and Object B - Object A is indicated by the red arrow pointing to the right side of the image. - Object B is indicated by the red arrow pointing to the left side of the image. 2. Determine the Position of Object A: - Object A is located on the right side of the image, near the black car. 3. Determine the Position of Object B: - Object B is located on the left side of the image, near the white truck. 4. Establish the First-Person Perspective of Object A: - If we imagine standing at the position of Object A, we need to consider the relative positions of other objects from this viewpoint. 5. Calculate the Direction of Object B from Object A: - From the position of Object A (right side), Object B (left side) would be to the left when looking straight ahead. 6. Consider the Orientation of the Scene: - The scene is a street view with a crosswalk. The direction "left" from Object A's perspective aligns with the left side of the image. Final Answer: From the first-person perspective of Object A, Object B is to the left.

🌀 **Step Partition & Judgment:**

✓ **step_type:** image description
**conclusion:** Object A is indicated by the red arrow pointing to the right side of the image

✓ **step_type:** image description
**conclusion:** Object B is indicated by the red arrow pointing to the left side of the image

✓ **step_type:** image description
**conclusion:** Object A is located on the right side of the image near the black car

✓ **step_type:** image description
**conclusion:** Object B is located on the left side of the image near the white truck

✓ **step_type:** logical inference
**conclusion:** Consider the relative positions of other objects from Object A's viewpoint

✓ **step_type:** logical inference
**conclusion:** From the position of Object A Object B would be to the left when looking straight ahead

✗ **step_type:** background information
**conclusion:** The scene is a street view with a crosswalk

✓ **step_type:** logical inference
**conclusion:** The direction left from Object A's perspective aligns with the left side of the image

### SRS = 7/8

**Figure 20** Examples of Step Relevance Score Evaluation.

**Reflection Validity Rate (RVR)**

**Question:** How can you speed up the separation of B's peels

**Answer:** Combined thermal and osmotic effects reduce peel adhesion strength

<image_1>

**Key Caption:**
(A) Thermos (B) orange
(C) baking soda (D) spoon...

📝 **Key Steps Annotation:**

1. 85C water flows from A to F  <image_2>

2. Granular diffusion occurs when D transfers C crystals into F  <image_3>

3. Osmotic pressure differential drives solution penetration into B's peel  <image_4>

---

**Single-image Input : <image_1>**

QVQ-72B:

redundant reflection

**Response :**

So I've got this problem here: "How to accelerate the separation of B peels?" And looking at the image, item B is an orange. So, I need to figure out how to speed up the process of separating the peel from the fruit. Let me think about this step by step. First, I need to understand what's involved in separating the peel from an orange. Normally, you'd take the orange and peel it manually, right? But that can be time-consuming and messy. So, maybe there's a better way to do it quickly. One idea that comes to mind is using a tool designed for peeling fruits. For example, there are citrus peelers that can help remove the peel efficiently. Maybe one of the other items in the image can be used for this purpose. Let's see: - A: A pink thermos bottle – Doesn't seem relevant for peeling an orange. - C: A small white bowl – Could be used for holding the peeled orange, but not for peeling. - D: An orange spoon – This might be useful for scooping out the flesh after peeling, but again, not directly for separating the peel. - E: A white spoon – Similar to D, probably not for peeling. - F: A glass mug – Not suitable for peeling. Hmm, so none of these items seem specifically designed for peeling oranges. Maybe I need to think of a different approach. Another thought: perhaps using heat or cold to make the peel easier to remove. For instance, microwaving the orange briefly could make the peel looser, allowing for easier separation. Alternatively, refrigerating it might have the opposite effect, making the peel tighter and harder to remove. So, maybe heat is a better option. But the image shows a thermos bottle, which is used for keeping drinks hot or cold. If I had to use that, perhaps filling it with hot water and letting the orange sit in warm water for a bit could soften the peel and make it easier to remove. However, I'm not sure if that's the most efficient way. Wait a minute, maybe I can use the thermos bottle to heat up the orange directly. But that seems unsafe and impractical. Let's discard that idea. Let me consider another angle. Perhaps cutting the orange with a knife could help in separating the peel. For example, scoring the peel and then lifting it off. But that seems like a standard peeling method and might not necessarily accelerate the process...

✴ **Step Partition & Judgment:**

...

✗ **conclusion:** None of the items are specifically designed for peeling oranges
 **error_type:** Wrong Conclusion

✗ **conclusion:** Using heat to make the peel easier to remove might be a better option
 **error_type:** Wrong Conclusion

✓ **conclusion:** Using the thermos bottle to heat up the orange directly seems unsafe and impractical
 **error_type:** N/A

✗ **conclusion:** Cutting the orange with a knife is a standard method and might not accelerate the process
 **error_type:** Wrong Conclusion
...

**RVR = 1/4**

**Figure 21** Examples of Reflection Validity Rate Evaluation.

# E LIMITATION

Our benchmark design, while aiming for rigorous evaluation of visual physical reasoning, inherits several limitations from both dataset construction and evaluation methodology. (i) Scene and Domain Coverage: Despite our efforts to include diverse physical scenarios, MVPBench cannot fully capture the long-tail distribution of real-world physics. This may limit the generalizability of conclusions drawn from our evaluation. To address this, we plan to iteratively expand the dataset with community feedback and new task paradigms. (ii) Annotation Subjectivity: Ground-truth reasoning chains, although carefully curated, may still carry annotator bias in step granularity or interpretation of visual cues. We mitigate this by introducing a graph-based CoT consistency metric to allow flexible yet principled comparisons across models. (iii) Model Usage Constraints: Our evaluation depends on the output of proprietary MLLMs (e.g., GPT-4o), which restricts full control over model internals

and fine-tuning procedures. As such, we treat model predictions as black-box outputs and encourage future work to validate findings across both open and closed-source systems for robustness.

# F    BROADER IMPACTS

**Positive Impacts:** On the positive side, this work has the potential to significantly enhance human-AI collaboration in fields such as education, scientific research, and accessibility, by enabling models to perform more transparent and interpretable reasoning across visual and textual modalities.

**Negative Impacts:** The potential negative societal impacts of our work are similar to those associated with other MLLMs and LLMs. The development of Visual CoT and MLLMs, while advancing AI, poses societal risks such as increased privacy invasion, the perpetuation of biases, the potential for misinformation, job displacement, and ethical concerns regarding accountability and consent.

**Mitigation Strategies:** To mitigate the aforementioned risks, several strategies are considered throughout the development and deployment of our model. First, we adopt a rigorous data curation process aimed at minimizing the propagation of harmful biases, ensuring that training data is as diverse, inclusive, and representative as possible. Second,privacy-preserving techniques such as data anonymization and adherence to data protection regulations (e.g., GDPR) are employed to safeguard user information. Third, we emphasize responsible release practices, including usage guidelines, model cards, and risk documentation, to inform users of the model's intended scope and limitations. Lastly, we advocate for continued interdisciplinary collaboration with ethicists, legal experts, and affected communities to ensure that the deployment of MLLMs aligns with broader societal values and norms.

# G    DETAILED EVALUATION PROMPTS

## G.1    COT QUALITY EVALUATION PROMPTS

## SAS Evaluation Prompt · 1

# Task Overview
Given a solution with multiple reasoning steps for an image-based problem, reformat it into well-structured steps and evaluate their correctness.

# Step 1: Reformatting the Solution
Convert the unstructured solution into distinct reasoning steps while:
- Preserving all original content and order
- Not adding new interpretations
- Not omitting any steps

## Step Types
1. Logical Inference Steps
   - Contains exactly one logical deduction
   - Must produce a new derived conclusion
   - Cannot be just a summary or observation

2. Image Description Steps
   - Pure visual observations
   - Only includes directly visible elements
   - No inferences or assumptions

3. Background Information Steps
   - External knowledge or question context
   - No inference process involved

## Step Requirements
- Each step must be atomic (one conclusion per step)
- No content duplication across steps
- Initial analysis counts as background information
- Final answer determination counts as logical inference

# Step 2: Evaluating Correctness
Evaluate each step against:

## Ground Truth Matching
For image descriptions:
- Key elements must match ground truth descriptions

For logical inferences:
- Conclusion must EXACTLY match or be DIRECTLY entailed by ground truth

## Reasonableness Check (if no direct match)
Step must:
- Premises must not contradict any ground truth or correct answer
- Logic is valid
- Conclusion must not contradict any ground truth
- Conclusion must support or be neutral to correct answer

## SAS Evaluation Prompt                                                    *2*

## Judgement Categories
- "Match": Aligns with ground truth
- "Reasonable": Valid but not in ground truth
- "Wrong": Invalid or contradictory
- "N/A": For background information steps

# Output Requirements
1. The output format MUST be in valid JSON format without ANY other content.
2. For highly repetitive patterns, output it as a single step.
3. Output maximum 35 steps. Always include the final step that contains the answer.

Here is the json output format:
## Output Format
[
  {{
    "step_type": "image description|logical inference|background information",
    "premise": "Evidence (only for logical inference)",
    "conclusion": "Step result",
    "judgment": "Match|Reasonable|Wrong|N/A"
  }}
]

Here is the problem, and the solution that needs to be reformatted to steps:

[Problem]

{question}

[Solution]

{solution}

[Correct Answer]

{answer}

[Ground Truth Information]

{gt_annotation}

## KSC Evaluation Prompt

You are an expert system for verifying solutions to image-based problems. Your task is to match the ground truth middle steps with the provided solution.

INPUT FORMAT:
1. Problem: The original question/task
2. A Solution of a model
3. Ground Truth: Essential steps required for a correct answer

MATCHING PROCESS:

You need to match each ground truth middle step with the solution:

Match Criteria:
- The middle step should exactly match in the content or is directly entailed by a certain content in the solution
- All the details must be matched, including the specific value and content
- You should judge all the middle steps for whethere there is a match in the solution

OUTPUT FORMAT:
JSON array of judgments:
[
  {{
    "step_index": <integer>,
    "judgment": "Matched" | "Unmatched",
  }}
]

ADDITIONAL RULES:
1. Only output the json array with no additional information.
2. Judge each ground truth middle step in order without omitting any step.

Here is the problem，answer, solution, and the ground truth middle steps:

[Problem]

{question}

[Answer]

{answer}

[Solution]

{solution}

[Ground Truth Information]

{gt_annotation}

## G.2 CoT Diversity Evaluation Prompts

## Diversity Evaluation Prompt

You are given a question about a physical experiment and several key reasoning steps.

Your goal is to identify ALL possible valid reasoning chains that logically connect the question to the final answer.

Each reasoning chain should include all key steps exactly once, arranged in a logically valid order.

Steps may be combined in different logical orders as long as the overall reasoning makes sense.

Think carefully: there may be multiple valid chains based on how the steps can be logically ordered.

Your job is to find as many valid logical chains as possible.
INPUT FORMAT:
1. Question: The original question/task
2. Final Answer: Answer to the original question
2. Key Reasoning Steps: A list of essential reasoning steps, each with an ID and explanation.

Output format

JSON array of judgments:
[
["key_step_1", "key_step_2", "key_step_3"],
["key_step_1", "key_step_3", "key_step_2"]
]

ADDITIONAL RULES:
1. Only output the json array with no additional information.

Here is the question, answer, and the Key Reasoning Steps:
[Question]

{question}

[Final Answer]

{answer}

[Solution]

{solution}

## G.3 COT EFFICIENCY EVALUATION PROMPTS

### PVR Rate  Prompt                                                          *1*

# Task Overview
Given a solution with multiple reasoning steps for an image-based problem, evaluate the relevance to get a solution (ignore correct or wrong) of each step.

# Step 1: Reformatting the Solution
Convert the unstructured solution into distinct reasoning steps while:
- Preserving all original content and order
- Not adding new interpretations
- Not omitting any steps

## Step Types
1. Logical Inference Steps
   - Contains exactly one logical deduction
   - Must produce a new derived conclusion
   - Cannot be just a summary or observation

2. Image Description Steps
   - Pure visual observations
   - Only includes directly visible elements
   - No inferences or assumptions

3. Background Information Steps
   - External knowledge or question context
   - No inference process involved

## Step Requirements
- Each step must be atomic (one conclusion per step)
- No content duplication across steps
- Initial analysis counts as background information
- Final answer determination counts as logical inference

# Step 2: Evaluating Relevancy
A relevant step is considered as: 75% content of the step must be related to trying to get a solution (ignore correct or wrong) to the question.

**IMPORTANT NOTE**:
Evaluate relevancy independent of correctness. As long as the step is trying to get to a solution, it is considered relevant. Logical fallacy, knowledge mistake, inconsistent with previous steps, or other mistakes do not affect relevance.
A logically wrong step can be relevant if the reasoning attempts to address the question.

## PVR Rate  Prompt                                                    *2*

The following behaviour is considered as relevant:
i. The step is planning, summarizing, thinking, verifying, calculating, or confirming an intermediate/final conclusion helpful to get a solution.
ii. The step is summarizing or reflecting on previously reached conclusion relevant to get a solution.
iii. Repeating the information in the question or give the final answer.
iv. A relevant image depiction shoule be in one of following situation: 1. help to obtain a conclusion helpful to solve the question later; 2. help to identify certain patterns in the image later; 3. directly contributes to the answer
v. Depicting or analyzing the options of the question is also relevant.
vi. Repeating previous relevant steps are also considered relevant.

The following behaviour is considered as irrelevant:
i. Depicting image information that does not related to what is asking in the question. Example: The question asks how many cars are present in all the images. If the step focuses on other visual elements like the road or building, the step is considered as irrelevant.
ii. Self-thought not related to what the question is asking.
iii. Other information that is tangential for answering the question.

# Output Format
[
  {{
    "step_type": "image description|logical inference|background information",
    "conclusion": "A brief summary of step result",
    "relevant": "Yes|No"
  }}
]

# Output Rules
Direct JSON output without any other output
Output at most 40 steps

Here is the problem, and the solution that needs to be reformatted to steps:

[Problem]

{question}

[Solution]

{solution}

## PCS Prompt

# Task
Evaluate reflection steps in image-based problem solutions, where reflections are self-corrections or
reconsiderations of previous statements.

# Reflection Step Identification
Reflections typically begin with phrases like:
- "But xxx"
- "Alternatively, xxx"
- "Maybe I should"
- "Let me double-check"
- "Wait xxx"
- "Perhaps xxx"
It will throw an doubt of its previously reached conclusion or raise a new thought.

# Evaluation Criteria
Correct reflections must:
1. Reach accurate conclusions aligned with ground truth
2. Use new insights to find the mistake of the previous conclusion or verify its correctness.

Invalid reflections include:
1. Repetition - Restating previous content or method without new insights
2. Wrong Conclusion - Reaching incorrect conclusions vs ground truth
3. Incompleteness - Proposing but not executing new analysis methods
4. Other - Additional error types

# Input Format
```
[Problem]
{question}

[Solution]
{solution}

[Ground Truth]
{gt_annotation}
```

# Output Requirements
1. The output format must be in valid JSON format without any other content.
2. Output maximum 30 reflection steps.

Here is the json output format:
## Output Format
```json
[
  {{
    "conclusion": "One-sentence summary of reflection outcome",
    "judgment": "Correct|Wrong",
    "error_type": "N/A|Repetition|Wrong Conclusion|Incompleteness|Other"
  }}
]
```
# Rules
1. Preserve original content and order
2. No new interpretations
3. Include ALL reflection steps
4. Empty list if no reflections found
5. Direct JSON output without any other output

## H  SETUP

### H.1  EXPERIMENT SETUP

**Evaluation Models.** To comprehensively assess performance on MVPBench, we selected a diverse array of multimodal large language models (MLLMs), encompassing both open-source and closed-source frameworks. Among open-source models, we evaluated LLaVA-OV 72BLi et al. (2025a), LLaVA-CoTXu et al. (2024), InternVL2.5 78BChen et al. (2024b), InternVL2.5-MPO 78BWang et al. (2024b), InternVL3 (78B, 78B-Instruct)Zhu et al. (2025), Qwen2.5-VL (7B, 72B)Bai et al. (2025), QVQ-72BQwen Team (2024), as well as the recently included Qwen2VL-2BWang et al. (2024a), MM Eureka-7BMeng et al. (2025), and R1-VL-2BZhang et al. (2025a), representing various architectures and multimodal integration strategies. Specifically, InternVL2.5-78B-MPO and InternVL3-78B-Instruct underwent mixed preference optimization (MPO) post-training, while InternVL2.5-78B and InternVL3-78B remained unmodified. Furthermore, Qwen2.5VL-7B and Qwen2VL-2B, along with their respective post-trained variants—MM Eureka-7B, which employs large-scale rule-based reinforcement learning (RL), and R1-VL-2B, utilizing Step-wise Group Relative Policy Optimization (StepGRPO)—are of significant interest. Additionally, prominent closed-source models such as GPT-4oOpenAI (2024), OpenAI o3OpenAI (2025), Claude 3.7 SonnetAnthropic (2025), Gemini-2.5Deepmind (2024), and Grok3xAI (2025) were selected based on their state-of-the-art multimodal reasoning capabilities. This expanded and carefully curated selection ensures a balanced and thorough evaluation encompassing both openly accessible and proprietary MLLM systems.

**Implementation Details.** All our experiments are conducted under a zero-shot setting, showcasing the generalization capacity of MLLMs for physical reasoning without few-shot prompting or further fine-tuning. By default, we employ the CoT prompting technique Wei et al. (2022), which encourages MLLMs to perform complete reasoning steps for fine-grained evaluation. All experiments are conducted on NVIDIA V100 GPUs.

### H.2  MODEL HYPERPARAMETERS

To ensure reproducibility and clarity regarding model settings used during evaluation, Table 10 provides detailed information on the hyperparameters and generation setups for each evaluated multimodal large language model (MLLM). Parameters not explicitly stated indicate that the default settings provided by the respective models were employed. This comprehensive specification facilitates transparent comparisons across models and experimental replication.

## I  THE USE OF LLMs

We employed large language models (LLMs) in a strictly auxiliary manner for (i) surface-level editing of the manuscript (grammar correction, minor rephrasing, and stylistic refinement), and (ii) technical assistance during dataset preparation, including checking the consistency of JSON schema, detecting formatting errors, and drafting preliminary scene descriptions for all curated datasets. All final annotations, dataset curation decisions, experimental designs, and analyses were exclusively performed and validated by the authors.

## J  REBUTTAL

**Table 10 Generating parameters for MLLMs.** Parameters not explicitly stated indicate the use of the model's default system settings.

| Model | Generation Setup |
|---|---|
| LLaVA-OV-72B | `torch.dtype=torch.float16, max_new_tokens=2048, temperature=0.7, device_map=balanced, min_pixels=256*28*28, max_pixels=768*28*28` |
| LLaVA-CoT | `torch.dtype=torch.float16, max_new_tokens=2048, temperature=0.7, device_map=balanced` |
| InternVL2.5-78B | `torch.dtype=torch.float16, max_new_tokens=2048, temperature=0.7, device_map=balanced_low_0` |
| InternVL2.5-78B-MPO | `torch.dtype=torch.float16, max_new_tokens=1024, temperature=0.7, device_map=balanced_low_0` |
| InternVL3-78B | `torch.dtype=torch.float16, max_new_tokens=1024, temperature=0.7, device_map=balanced_low_0` |
| InternVL3-78B-Instruct | `torch.dtype=torch.float16, max_new_tokens=1024, do_sample=False, temperature=0.7, device_map=balanced_low_0` |
| Qwen2.5-VL-7B | `torch.dtype=torch.float16, max_new_tokens=1024, do_sample=False, temperature=0.7, device_map=balanced` |
| Qwen2.5-VL-72B | `torch.dtype=torch.bfloat16, temperature=0.7, max_new_tokens=1024, device_map=balanced, min_pixels=256*28*28, max_pixels=768*28*28` |
| QVQ-72B | `torch.dtype=torch.float16, max_new_tokens=512, do_sample=False, temperature=0.7, min_pixels=256*28*28, max_pixels=768*28*28, device_map=balanced,` |
| MM-Eureka-7B | `torch.dtype=torch.float16, max_new_tokens=2048, do_sample=False, temperature=0.7, device_map=balanced` |
| Qwen2VL-2B | `torch.dtype=torch.bfloat16, max_new_tokens=2048, do_sample=False, temperature=0.7, device_map=balanced` |
| R1-VL-2B | `torch.dtype=torch.float16, max_new_tokens=2048, use_cache=True, temperature=0.7` |
| GPT-4o | `dtype=torch.float16, sampling=False, temperature=0.2, max_new_tokens=1024` |
| OpenAI o3 | `dtype=torch.float16, sampling=False, temperature=0.2, max_new_tokens=1024` |
| Claude 3.7 Sonnet | `dtype=torch.float16, sampling=False, temperature=0.2, max_new_tokens=1024` |
| Gemini-2.5-flash-preview-04-17 | `dtype=torch.float16, sampling=False, temperature=0.2, max_new_tokens=1024` |
| Grok3 | `dtype=torch.float32, sampling=False, temperature=0.2, max_new_tokens=1024` |

ChatGPT 4o ⌄

In the figure, the arrows point to object A. What is the direction of movement for object A?Please answer the question step by step.

ChatGPT 4o ⌄

To determine the direction of movement for object A (the orange vehicle) in the image, let's go through the reasoning step by step:

---

**Step 1: Understand the Road Structure**
- The image shows a curved, two-lane road with lane markings.
- There is a solid yellow line near the center, which typically separates opposing traffic directions

**Step 2: Observe Vehicle Placement**
- The vehicle labeled as object **A** is on the right side of the solid yellow line (from the vehicle's perspective), which usually indicates it is in its designated lane, assuming right-hand traffic rules (as in most countries).
- The dashed white line to the left of the vehicle typically separates two lanes of traffic moving in the **same** direction.
- This implies a **multi-lane one-way road** OR the vehicle is about to enter a **two-way section** (less likely due to the curve and markings).

---

**Step 3: Direction of Curvature**
- The road curves to the **left** from the driver's perspective.
- The vehicle appears to be moving **towards the top of the image**, as the front of the vehicle (windshield, driver area) is pointed that way.
- The shadowing and motion blur also support this direction.

---

**Step 4: Arrow and Label Interpretation**
- The red arrow points **from the front of the vehicle** (object A) toward the left side of the image.
- This is not the direction of movement but likely used just to label the object as A.

---

✅ **Conclusion**

**Object A (the vehicle) is moving forward along the road, which is towards the top-left corner of the image.**

**Figure 22** **Example of CoTS with GPT-4o.**

**Table 11 Expanded Subcategory-level Evaluation of CoT Reasoning in MVPBench: Human Baselines.**
We present a detailed subcategory-level evaluation of CoT reasoning along the dimensions of *Quality*, *Diversity*, and *Efficiency*.

| Human Performance | Phys-Experiment | | | Phys-Problems | | | Spatial-Relation | | | Dyn-Prediction | | |
|---|---|---|---|---|---|---|---|---|---|---|---|---|
| | Quality | Diversity | Efficiency | Quality | Diversity | Efficiency | Quality | Diversity | Efficiency | Quality | Diversity | Efficiency |
| | 94.31 | 98.72 | 100.00 | 93.28 | 96.42 | 99.12 | 100.00 | 99.13 | 100.00 | 99.82 | 95.76 | 100.00 |

**Table 12 Sensitivity analysis of the weighting hyperparameters.** Top-performing models within each category are highlighted in blue (open-source) and red (closed-source).

| Model | Phys-Experiment | | | Phys-Problems | | | Spatial-Relation | | | Dyn-Prediction | | |
|---|---|---|---|---|---|---|---|---|---|---|---|---|
| | $\gamma=0.8$ | $\gamma=0.5$ | $\gamma=0.2$ | $\gamma=0.8$ | $\gamma=0.5$ | $\gamma=0.2$ | $\gamma=0.8$ | $\gamma=0.5$ | $\gamma=0.2$ | $\gamma=0.8$ | $\gamma=0.5$ | $\gamma=0.2$ |
| *Open-source MLLMs* | | | | | | | | | | | | |
| LLaVA-OV-72B Li et al. (2025a) | 53.27 | 66.61 | 79.65 | 76.68 | 79.72 | 82.18 | 46.31 | 59.79 | 72.75 | 85.37 | 89.93 | 93.22 |
| LLaVA-CoT Xu et al. (2024) | 40.41 | 52.34 | 63.35 | 32.15 | 45.46 | 58.28 | 36.81 | 54.31 | 72.54 | 20.14 | 41.61 | 61.52 |
| InternVL2.5-78B Chen et al. (2024b) | 63.02 | 73.38 | 84.15 | 63.34 | 71.32 | 79.52 | 63.71 | 71.08 | 79.13 | 83.57 | 87.49 | 91.02 |
| InternVL2.5-78B-MPO Wang et al. (2024b) | 72.27 | 79.43 | 87.39 | 73.16 | 75.87 | 78.29 | 64.21 | 71.42 | 78.51 | 87.05 | 90.60 | 93.67 |
| InternVL3-78B Zhu et al. (2025) | 73.14 | 83.39 | 93.49 | 61.26 | 68.23 | 75.52 | 63.32 | 70.43 | 77.12 | 84.48 | 88.05 | 92.05 |
| InternVL3-78B-Instruct Zhu et al. (2025) | 65.16 | 74.57 | 83.23 | 62.32 | 69.79 | 76.24 | 64.15 | 70.96 | 77.48 | 85.77 | 89.58 | 93.26 |
| Qwen2.5-VL-7B Bai et al. (2025) | 75.42 | 80.15 | 86.43 | 65.71 | 67.10 | 69.41 | 51.32 | 68.57 | 85.36 | 78.41 | 82.16 | 86.33 |
| Qwen2.5-VL-72B Bai et al. (2025) | 77.53 | 82.59 | 87.72 | 71.39 | 79.69 | 87.52 | 45.61 | 59.67 | 72.64 | 96.12 | 98.75 | 99.85 |
| QVQ-72B Qwen Team (2024) | 0.00 | 0.00 | 0.00 | 0.00 | 0.00 | 0.00 | 0.00 | 0.00 | 0.00 | 0.00 | 0.00 | 0.00 |
| *Closed-source MLLMs* | | | | | | | | | | | | |
| GPT-4o OpenAI (2024) | 62.28 | 76.36 | 91.01 | 62.72 | 69.52 | 77.32 | 56.46 | 69.33 | 81.27 | 89.57 | 91.14 | 93.21 |
| OpenAI o3 OpenAI (2025) | 68.52 | 75.58 | 82.43 | 62.76 | 68.57 | 75.41 | 65.72 | 71.26 | 77.41 | 90.41 | 91.18 | 92.61 |
| Claude 3.7 Sonnet Anthropic (2025) | 75.43 | 78.97 | 82.05 | 68.07 | 72.15 | 76.14 | 59.31 | 72.71 | 85.72 | 92.03 | 92.47 | 93.54 |

**Table 13 Evaluated Results on unbalanced dataset** Top-performing models within each category are highlighted in blue (open-source) and red (closed-source).

| Model | Phys-Experiment | | | Phys-Problems | | | Spatial-Relation | | | Dyn-Prediction | | |
|---|---|---|---|---|---|---|---|---|---|---|---|---|
| | Quality | Diversity | Efficiency | Quality | Diversity | Efficiency | Quality | Diversity | Efficiency | Quality | Diversity | Efficiency |
| *Open-source MLLMs* | | | | | | | | | | | | |
| LLaVA-OV-72B Li et al. (2025a) | 38.32 | 67.83 | 97.31 | 32.98 | 80.32 | 99.01 | 37.32 | 58.32 | 99.12 | 41.66 | 89.93 | 99.72 |
| LLaVA-CoT Xu et al. (2024) | 34.41 | 52.68 | 97.23 | 21.14 | 45.78 | 98.76 | 33.21 | 54.21 | 99.02 | 43.77 | 41.61 | 99.78 |
| InternVL2.5-78B Chen et al. (2024b) | 44.61 | 73.32 | 94.67 | 47.48 | 73.41 | 98.87 | 44.29 | 70.03 | 99.53 | 44.78 | 87.49 | 100 |
| InternVL2.5-78B-MPO Wang et al. (2024b) | 41.75 | 79.98 | 97.42 | 52.69 | 76.18 | 99.18 | 40.52 | 72.01 | 96.12 | 44.06 | 90.60 | 99.76 |
| InternVL3-78B Zhu et al. (2025) | 38.31 | 84.52 | 92.01 | 58.32 | 68.91 | 98.95 | 43.98 | 73.65 | 98.26 | 46.68 | 88.05 | 99.95 |
| InternVL3-78B-Instruct Zhu et al. (2025) | 44.02 | 74.66 | 94.82 | 52.65 | 69.80 | 99.79 | 42.01 | 74.86 | 97.97 | 44.20 | 89.58 | 99.96 |
| Qwen2.5-VL-7B Bai et al. (2025) | 37.66 | 81.02 | 96.77 | 42.76 | 67.80 | 98.45 | 36.31 | 67.87 | 96.01 | 40.30 | 82.16 | 99.85 |
| Qwen2.5-VL-72B Bai et al. (2025) | 41.32 | 83.21 | 96.54 | 57.32 | 80.04 | 99.32 | 40.12 | 60.97 | 98.12 | 46.94 | 98.75 | 99.65 |
| QVQ-72B Qwen Team (2024) | 51.32 | 0.00 | 72.54 | 60.86 | 0.00 | 65.31 | 40.15 | 0.00 | 70.41 | 69.20 | 0.00 | 79.13 |
| *Closed-source MLLMs* | | | | | | | | | | | | |
| GPT-4o OpenAI (2024) | 51.73 | 76.63 | 97.34 | 53.55 | 69.97 | 99.03 | 48.31 | 68.32 | 99.82 | 52.35 | 91.14 | 99.59 |
| OpenAI o3 OpenAI (2025) | 58.36 | 76.22 | 97.53 | 66.30 | 69.27 | 99.05 | 46.19 | 73.15 | 99.42 | 69.44 | 91.18 | 99.71 |
| Claude 3.7 Sonnet Anthropic (2025) | 49.45 | 79.52 | 97.39 | 57.98 | 74.51 | 93.12 | 46.10 | 73.76 | 99.87 | 54.92 | 92.47 | 98.45 |

