# OpenReview forum: "Seeing is Not Reasoning: MVPBench for Graph-based Evaluation of Multi-path Visual Physical CoT"
_ICLR.cc/2026/Conference — Submitted to ICLR 2026_

### Official Review · Reviewer_xjhL · 2025-10-26

**Soundness:** 3
**Presentation:** 3
**Contribution:** 2
**Rating:** 4
**Confidence:** 4

**Summary:**

This paper tries to address the problem: even state-of-the-art MLLMs like GPT-4o and O3 fail at visual physical reasoning. To address the shortcomings of existing benchmarks (e.g., unrealistic data, text shortcuts), the authors propose MVPBench, a new benchmark based on real-world physics scenarios featuring multi-image inputs (visual CoT) and multi-path correct answers. Furthermore, the paper introduces a graph-based evaluation suite to assess the process of reasoning across quality, diversity, and efficiency. A key finding is that RL-based alignment post-training (like MPO) can actively harm the model's spatial reasoning abilities.

**Strengths:**

1. The target issues of the paper are meaningful and worth exploring and the motivation is clear.
2. The paper is well written and easy to follow.
3. The design of MVPBench directly targets the weaknesses of current benchmarks, forcing models to ground their reasoning in dynamic visual evidence rather than relying on linguistic priors.
4. The discovery that RL post-training impairs physical reasoning is a significant and non-obvious contribution that has important implications for future MLLM alignment strategies.

**Weaknesses:**

1. The new graph-based CoT evaluation metric (Sec 4.2) is excessively complex. It relies on Graph Edit Distance (GED) and multiple hand-tuned hyperparameters, making the evaluation difficult to reproduce and its superiority over simpler metrics unproven.
2. Key metrics for CoT Quality and Efficiency depend on GPT-4o as the judge. This is highly circular, as GPT-4o is one of the models being evaluated, and this method may systematically favor models with a similar style to GPT-4o.
3. The benchmark's scale is relatively small (1,211 samples). More importantly, the data is scraped from sources like Bilibili and Gaokao exams (Sec 3.1), raising concerns about annotation quality and noise that are not discussed.
4. The "multi-path CoT" is a core feature, but this annotation is extremely expensive and subjective (how to define "all" valid paths?). The paper does not report Inter-Annotator Agreement for this process, making the reliability and scalability of this feature questionable.

**Questions:**

1. Line 988, Chinses -> Chinese
2. The claim that "RL post-training... harms spatial reasoning" is very strong. The evidence is limited to a few models, which may be an artifact of specific RL implementations (like MPO) rather than a fundamental flaw of RL itself. Can authors discuss more about it?
3. The human performance evaluation is very weak. It only assesses the "diversity" metric, and the subjects were given the key steps, which is not comparable to the model's task of generating CoT from scratch. This makes the human scores in Table 5 not very meaningful.
4. See weakness

---

> ### Author Response · Authors · 2025-11-21
>
> ### **Q1. Typographical Error**
>
> Thank you for your careful observation. We have corrected the typo from “Chinses” to “Chinese” at line 988.
>
> ### **Q2. The claim that "RL post-training... harms spatial reasoning**
>
> Thank you for pointing this out. Our intention was not to claim that RL post-training inherently harms spatial reasoning, but to report that post-training may reduce generalization on our visual physical benchmark compared to the corresponding base models. To avoid potential over-interpretation, in the revised version we will explicitly state that, on our benchmark and under the specific post-training setups we tested, some post-trained models show weaker spatial generalization than their base versions.
>
> Importantly, **this observation is not based on a single RL algorithm**. As detailed in Appendix A.2, we compare four pairs of models: InternVL2.5 and InternVL3-instruct (no post-training) vs. InternVL2.5-MPO and InternVL3 (**trained with MPO**), and Qwen2.5VL-7B and Qwen2VL-2B (no post-training) vs. MM Eureka-7B (**trained via large-scale rule-based RL**) and R1-VL-2B (**trained via Step-wise Group Relative Policy Optimization**). **Across these diverse RL strategies, the post-trained variants consistently exhibit lower step-level accuracy, path validity, and path coverage (Figures 5–6), even though some alignment-oriented subset sometimes improve**. A plausible interpretation is that many existing post-training pipelines optimize for reward data and objectives that underweight visual-centric spatial reasoning, which may introduce distributional bias or over-optimization toward non-visual preferences. We will make this limitation explicit and frame our claim more carefully as a hypothesis grounded in the observed model families.
>
> ### **Q3. The human performance evaluation only assesses the "diversity" metric**
>
> Thank you for your feedback. We acknowledge that human evaluation in multimodal CoT generation is inherently challenging, as humans cannot be reasonably expected to produce long reasoning chains “from scratch” in a controlled and reproducible manner. This limitation motivated our focus on the **diversity** metric in the initial human study.
>
> While MVPBench is framed as a visual question answering (VQA) benchmark, its evaluation protocol requires detailed, step-by-step reasoning chains, making the **quality** and **efficiency** metrics difficult to assess reliably for human participants. To provide a more complete comparison, we conducted an additional human study: we sampled 15 instances from each of the four major subsets and asked participants to generate full reasoning chains under the same constraints applied to model evaluation.
>
> The results reveal clear differences across subsets. Humans achieve near-perfect performance on Spatial Relations and Dynamic Prediction, reflecting strong alignment with intuitive physical reasoning. Performance drops on Physics Experiments and Physics Problems, which demand higher conceptual understanding and more complex procedural reasoning. The detailed results are provided in **Appendix J, Table 11**.
>
> We hope this clarifies our human baseline evaluation setup.
>
> ### **Q4. Concerns about annotation quality, the benchmark's scale and noise**
>
> Thank you for raising this point. We agree that, compared to large-scale web benchmarks, our dataset size (1,211 examples) is relatively modest. This is partly a deliberate design choice: each item in our benchmark is not a single QA pair, but a **multi-step, multi-path visual physical CoT example**, with detailed step-level and path-level annotations. As a result, the amount of *reasoning supervision* per item is substantially higher than in standard QA datasets, and each example is expensive to construct. Our focus in this work is on **high-quality, diagnostic physical reasoning trajectories** rather than sheer volume.
> Regarding data sources, we would like to clarify that while the **raw data** (videos, images, exam-style problems) are collected from platforms such as Bilibili and Gaokao-style exams, **all labels and CoTs are manually created and verified by our annotation team** after scraped. As detailed in Sec. 3.1 / Appendix C, we employ a multi-stage pipeline: (i) initial filtering to remove low-quality, ambiguous, or non-physical content; (ii) expert-designed templates for questions and options; (iii) manual construction of human CoTs (with multiple valid reasoning paths); and (iv) **double annotation and cross-checking** for every item, with disagreements resolved by more experienced annotators. In the revised version, we will expand Sec. 3.1 and Appendix C to explicitly discuss potential noise sources, describe the filtering criteria, and add quantitative statistics on annotation checks, so that the annotation quality and noise control are more transparent.
> We agree that scaling up to larger, more diverse physical settings is an important next step.

---

> > ### Comment · Reviewer_xjhL · 2025-11-27
> >
> > I sincerely appreciate the authors for their response. However, my concern in weakness 1 and weakness 2 remain unsolved.

---

> ### Author Response · Authors · 2025-11-27
>
> We sincerely thank the reviewer for the follow-up comment. Due to space limitations in our initial rebuttal, we were unable to fully address Weakness 1 and Weakness 2. We provide a complete clarification below.
>
> ### **W1: The graph-based CoT evaluation metric is overly complex and relies on hand-tuned hyperparameters.**
>
> #### **(1) Why GED is necessary — limitations of the simpler DAG-based matching**
>
> Our initial design indeed adopted a **pure DAG-based matching framework**. However, extensive experiments revealed a critical limitation:
> **DAG matching is overly sensitive to minor variations in node/edge topology.**
> Even when models produce *logically correct* and *human-plausible* intermediate steps, slight reordering or the addition of equivalent branches is treated as a hard mismatch. Empirically, this leads to the following issues:
>
> * **Systematically lower scores** that underestimate models’ true reasoning quality
> * **Artificially inflated gaps between models**, as different models are penalized unevenly
> * **Unreliable cross-model comparison**, since the strict topology match fails to reflect genuine performance differences
>
> These issues are clearly visible in the table below.
>
> |Model|Phys-Experiment| |Phys-Problems| |Spatial-Relation| |Dyn-Prediction| |
> |-----|----------------|-|-------------|-|-----------------| -|--------------|-|
> |     |DAG-based|With GED|DAG-based|With GED|DAG-based|With GED|DAG-based|With GED|
> |**Open-source MLLMs**|||||||||
> |LLaVA-OV-72B|56.34|10.27↑|75.54|4.18↑|49.32|10.47↑|86.72|3.21↑|
> |LLaVA-CoT|39.41|12.93↑|34.72|10.74↑|39.67|14.64↑|25.13|16.48↑|
> |InternVL2.5-78B|65.06|8.32↑|64.27|7.05↑|62.29|8.79↑|83.26|4.23↑|
> |InternVL2.5-78B-MPO|73.17|6.26↑|72.69|3.18↑|62.21|9.21↑|87.32|3.28↑|
> |InternVL3-78B|74.66|8.73↑|62.32|5.91↑|61.29|9.14↑|83.88|4.17↑|
> |InternVL3-78B-Instruct|66.91|7.66↑|64.50|5.29↑|61.89|9.07↑|85.94|3.64↑|
> |Qwen2.5-VL-7B|74.66|5.49↑|63.30|3.80↑|54.70|13.87↑|78.50|3.66↑|
> |Qwen2.5-VL-72B|77.15|5.44↑|73.82|5.87↑|47.70|11.97↑|98.32|0.43↑|
> |**Closed-source MLLMs**|||||||||
> |GPT-4o|65.73|10.63↑|63.55|5.97↑|60.17|9.16↑|88.76|2.38↑|
> |OpenAI o3|68.36|7.22↑|64.30|4.27↑|66.11|5.15↑|87.75|3.43↑|
> |Claude 3.7 Sonnet|74.20|4.77↑|68.53|3.62↑|61.96|10.75↑|92.29|0.18↑|
>
> * ↑ Indicates performance improvement after integrating GED.
>
> Therefore, we integrate Graph Edit Distance (GED) into the DAG-based framework to reduce topology sensitivity and allow more tolerant comparison of structurally variant yet valid reasoning paths.
>
> #### **(2) On reproducibility and hyperparameter sensitivity**
>
> We conducted a **systematic sensitivity analysis** on the key hyperparameter **γ**, which converts GED into a smooth similarity score. The complete ablation results are provided in **Appendix J, Table 12**, with further discussion in our response to reviewer *t9yx*. The results consistently show that the metric remains **stable across a wide range of γ**, and the relative ranking of models does not change.
>
> This demonstrates that the metric’s behavior is stable, and its conclusions do not depend on fragile or heavily tuned choices. Moreover, all other involved parameters follow **standard settings adopted in prior work**, using default fusion weights of **0.7 and 0.3**, ensuring reproducibility and avoiding additional tuning.
>
> ### **W2: This method may systematically favor models with a similar style to GPT-4o.**
> Thank you for raising this concern. We address it from both an empirical and a methodological perspective.
>
> (1) Empirically, GPT-4o as judge does not systematically favor GPT-style models.
> In the updated tables we have added the results of Gemini-2.5-pro and GPT-5 (thinking).
>
> |Model|Phys-Experiment| | |Phys-Problems| | |Spatial-Relation| | |Dyn-Prediction| | |
> |-----|--------------|-|-|--------------|-|-|----------------|-|-|----------------|-|-|
> |     |Quality|Diversity|Efficiency|Quality|Diversity|Efficiency|Quality|Diversity|Efficiency|Quality|Diversity|Efficiency|
> |Gemini-2.5-pro|62.69|80.00|100.00|73.44|57.68|98.33|51.91|55.61|100.00|35.89|66.67|99.39|
> |GPT-5|56.36|75.64|99.41|69.56|69.86|88.97|44.56|44.07|99.17|48.17|59.58|96.19|
>
> As shown there, **Gemini-2.5-pro scores higher CoT Quality and Efficiency than GPT-5 on 3 of 4 sub-benchmarks (with 100% efficiency on two), despite a very different CoT style. Several non-OpenAI models (e.g., Qwen, InternVL) also achieve the best scores in some Table 3 cells under a GPT-4o judge**,  indicating that our evaluation favors correctness and step coverage rather than “GPT-like” wording.
>
> (2) Before fixing the judge, we piloted Qwen-VL-7B/32B, GPT-4o and GPT-o1 as judges. GPT-o1 aligned best with humans, but GPT-4o was only slightly worse yet ~10× cheaper/faster, so we chose it as a frozen judge. Moreover, using strong LLMs as judges (e.g., MT-Bench) now is standard. In terms of method design, the judge only compares candidate CoTs to human key steps (no model ID), and metrics use these steps plus normalized length, reducing style bias.

---

### Official Review · Reviewer_t9yx · 2025-10-31

**Soundness:** 2
**Presentation:** 3
**Contribution:** 2
**Rating:** 4
**Confidence:** 4

**Summary:**

The paper presents MVPBench, a benchmark designed to evaluate the visual physical reasoning abilities of multimodal large language models (MLLMs). It contains 1,211 examples across physics experiments, physics problems, spatial relations, and dynamic prediction, each annotated with multiple valid chain-of-thought (CoT) paths. To evaluate the reasoning process, the authors propose a graph-based CoT evaluation framework from three dimensions: quality, diversity, and efficiency. Experiments show that multi-image inputs improve reasoning, while reinforcement learning–based post-training surprisingly reduces general reasoning performance, indicating the misalignment between fine-tuning and genuine physical understanding.

**Strengths:**

- The paper jointly assesses CoT quality, diversity, and efficiency, enabling a fine-grained measurement of model behavior.
- MVPBench provides multiple annotated reasoning paths for each question. This design better reflects real-world human problem-solving patterns and allows graph-based metrics.
- The benchmark removes the textual cues to force the model to use visual evidence for reasoning. This helps reduce the text priors as a shortcut and isolates the visual reasoning ability.

**Weaknesses:**

- Figure 1 incorrectly labels GPT-4o’s Step 1 as wrong. However, step 1 in textual CoT is wrong but is considered correct.
- The first subsection in the related work, “Limitations of Multi-modal Large Language Models”, is too broad and unfocused. The authors only discuss the limitations of physical understanding.
- The same symbol \alpha is reused in both CRS and Path Coverage Scores without clear differentiation. Moreover, the paper first uses “DAG-based matching” (L299–300) before defining.
- Although multi-image input improves performance for most models, the paper does not analyze the reasons. Besides, why the QVQ-72B model uniquely degrades with more images still remains unclear.
- The weighting coefficient hyperparameters are fixed without sensitivity testing. This limits the confidence in metric stability across different setups.
- The claim that RL post-training harms spatial reasoning is attributed to distributional bias or overfitting, but no empirical evidence is presented to support this hypothesis.
- The dataset is unbalanced. It consists of only 100 Dynamic Prediction problems. Additionally, half of the Physics Experiments are Mechanics problems. This may bias performance averages and limit conclusions on underrepresented subdomains.
- The SAS and SRS metrics rely heavily on GPT-4o for evaluation, which not only requires high computational and financial cost but may also introduce evaluation bias, potentially affecting the reliability of the assessment.

**Questions:**

- Could the authors correct Figure 1 and clarify why GPT-4o’s reasoning is considered wrong? Also, please ensure consistent weight notation and define DAG before its first use.
- Could the authors refine the first subsection in Related Work on a more specific scope?
- Could the authors analyze the reasons for performance improvement from multi-image inputs and the degradation in QVQ-72B with empirical evidence?
- Could the authors conduct experiments to confirm the cause of the generalizability reduction of RL post-training in physical visual reasoning?
- Could the authors include a sensitivity analysis of the weighting hyperparameters to assess the robustness of the proposed metrics?
- Given the unbalanced dataset, could the authors evaluate whether the performance remains consistent when balanced subsets are sampled?

---

> ### Author Response · Authors · 2025-11-21
>
> ### **Q1. Figure 1**
> Thank you for pointing this out. In the original Figure 1, the bus orientation was misrepresented: the Step-1 human CoT incorrectly stated that “the tail is near the bottom,” which led to a mismarked reasoning step.
>
> This example corresponds to item #72 in the Spatial Relations subset. The original benchmark question—*“the front of A is facing toward the lower, and the rear toward the upper”*—is correct in the dataset. The issue arose solely from the simplified schematic used for Figure 1, which was intended as an explanatory illustration rather than the actual benchmark item.
>
> In the revised PDF, we have redrawn the figure with the correct orientation, updated the human Step-1 CoT, corrected the model-step annotation, and added the original GPT-4o output in the appendix (Fig. 22) for completeness.
>
> This correction is limited to the figure and **does not affect any dataset items or experimental results**.
>
> ### **Q2. Related Works**
> Thank you for the suggestion. Our intention in this subsection was not to discuss the global limitations of MLLMs, but specifically to summarize prior work on visual physical and spatial reasoning. We agree that the current title “Limitations of Multi-modal Large Language Models” is broader than the actual scope and can be misleading.
> In the revised pdf, we refined the scope explicitly by renaming the subsection to “Limitations of MLLMs in Visual Physical Reasoning”, and slightly rewriting the opening sentence to make clear that we focus on physical understanding from visual input, rather than general MLLM limitations.
>
> ### **Q3. Evidence for Multi-Image Benefits and QVQ-72B Degradation**
> Thank you for the suggestion. We have conducted a detailed empirical analysis of both phenomena, and the evidence is presented in Appendix D of our paper. Below is a brief summary.
>
> - Why Multi-Image Inputs Improve Performance
>
>   As shown in Appendix D (Figures 11–19), our side-by-side comparisons of single-image vs. multi-image inputs reveal that:
>
>     1. Richer cross-image reasoning: Multi-image inputs allow the model to compare, track changes, and combine complementary visual cues, leading to more complete and coherent reasoning.
>     2. Direct metric gains: This improved reasoning capability results in notable improvements on SAS–KSC and PVR–PCS, which evaluate semantic accuracy and physical reasoning depth.
>
> - Why QVQ-72B Performance Degrades
>
>   Appendix D.3 (Figure 21) shows two main causes:
>
>     1. Excessive reflective text: QVQ-72B frequently produces long, repetitive meta-reasoning, diluting the core answer and reducing factual precision.
>     2. Token-limit truncation: Its verbosity causes it to hit the shared token limit earlier than other models, leading to truncated or incomplete answers and further score degradation.
>
> ### **Q4. On the Generalizability Reduction of RL Post-Training**
>
> Thank you for this thoughtful question. Our explanation—that RL post-training may introduce distributional bias or overfitting to the post-training distribution—is a hypothesis, motivated by consistent patterns we observe across four model families trained with different RL strategies, rather than a fully established causal mechanism. As this issue overlaps with concerns raised by Reviewer xjhL, we refer the reviewer to our response to Reviewer xjhL (Q2). A full causal investigation is beyond the scope of this benchmark paper.
>
> ### **Q5. Sensitivity Analysis of Weighting Hyperparameters**
> Thank you for the suggestion. We have conducted a sensitivity analysis on the key hyperparameter (gamma), which is used to convert the GED distance into a smooth similarity score. Specifically, we tested three values: **0.2**, **0.5**, and **0.8**.
>
> We observed that:
>
> * (gamma = 0.2) yields consistently higher overall similarity scores, but it also **reduces the distinguishability** between different models, making it difficult to capture meaningful performance gaps.
> * (gamma = 0.8) provides more separation than 0.2, but the metric variability is still smaller compared to our default setting.
> * (gamma = 0.5)—the value used in the main paper—strikes the best balance, offering **stable performance** while preserving **clear differences across models**.
>
> Based on these results, we consider (gamma = 0.5) to be a robust and reasonable choice. The full ablation results are provided in **Appendix J, Table 12**.
>
> ### **Q6. Evaluated Unbalanced Dataset**
> We appreciate the reviewer’s suggestion. To assess robustness under balanced sampling, we uniformly sampled 100 instances from each of the three subcategories in *Physics Experiments → Physics Problems → Spatial Relations* and repeated the evaluation. Results are provided in **Appendix J, Table 13**.
>
> Quality and Diversity show moderate variation, whereas Efficiency remains highly stable. Crucially, the performance trends on this balanced subset are consistent with those from the full dataset, indicating that our conclusions are not driven by data imbalance.

---

### Official Review · Reviewer_nCwg · 2025-11-02

**Soundness:** 3
**Presentation:** 3
**Contribution:** 2
**Rating:** 6
**Confidence:** 4

**Summary:**

This paper examines the limitations of multimodal large language models (MLLMs) in performing visual physical reasoning—understanding and reasoning about motion, spatial relations, and causality from visual inputs. Despite recent advancements exemplified by models such as OpenAI o3 and GPT-4o, the study finds that these systems exhibit fundamental weaknesses in applying physical laws and maintaining coherent, visually grounded reasoning chains, particularly across multi-step inference tasks. To address this gap, the authors introduce MVPBench, a benchmark designed to evaluate visual chain-of-thought (CoT) reasoning using multi-image inputs that require temporally grounded, step-by-step reasoning. A graph-based CoT consistency metric is proposed to assess the logical validity of models’ reasoning processes while mitigating reliance on textual shortcuts. Experimental evaluations reveal that even state-of-the-art MLLMs demonstrate poor performance in visual physical reasoning and weak image–text alignment, and that reinforcement learning–based alignment methods may inadvertently impair spatial reasoning. These findings underscore the need to revisit current fine-tuning practices for multimodal reasoning models.

**Strengths:**

- The introduction of MVPBench, a new benchmark for evaluating visual physical reasoning with multi-image inputs and a graph-based consistency metric, provides a rigorous framework for assessing multimodal models’ reasoning capabilities.
- The paper offers valuable insights into the current weaknesses of state-of-the-art models in visual physical reasoning, particularly in understanding basic physical laws and spatial interactions, and challenges existing practices in reinforcement learning-based alignment.

**Weaknesses:**

- In Table 3, it appears that closed-source LLMs generally outperform their open-source counterparts. This raises the question of whether the advanced reasoning capabilities of new models are already (or partially) addressing the tasks in question. To investigate, I reviewed the example in Figure 1 for both GPT-5 (thinking) and Gemini-Pro 2.5, and found that both models were able to correctly answer the question. Notably, Gemini-Pro 2.5 even demonstrated the ability to correctly identify the movement trajectory in the image.
- Some insights presented in the paper seem to echo findings from previous studies. For example, the claim that "Providing models with the full image sequence boosts performance by up to 21% points-evidence that temporal context matters" partially aligns with "thinking with images." that once models are capable of processing multiple images, leveraging more images for reasoning should enhance performance across many multimodal tasks.
- Given the title of the submission, "Visual Physical CoT", I had anticipated a broader exploration of "physical content" beyond simple physics experiments. For instance, it would be valuable to see reasoning applied to more complex dynamic systems, such as those found in earth science (e.g., weather prediction), or other scientific domains, where physical reasoning plays a crucial role in understanding complex phenomena.

**Questions:**

Please refer to the weaknesses.

---

> ### Author Response · Authors · 2025-11-21
>
> ### **Q1. Whether the advanced reasoning capabilities of new models are already (or partially) addressing the tasks in question**
>
> We appreciate your observation that the closed-source models in Table 3 generally outperform open-source models, and that more recent systems such as GPT-5 (thinking) and Gemini-2.5-pro can correctly solve the specific Figure 1 example (including identifying the correct motion trajectory). **Figure 1 is intended as a didactic illustration rather than a representative or particularly hard item, so it is expected that strong models can handle this case**. Our main claims are instead based on the full benchmark results and diagnostic breakdowns, where even the strongest models still fall substantially short of human performance and show flaws in tracking object states across multiple events, reasoning about occluded or implicit interactions, and maintaining consistency between visual and textual reasoning.
>
> In response to your suggestion, **we have added GPT-5 (thinking) and Gemini-2.5-pro to the following table**. As shown in the new results, both models achieve high generation efficiency and stronger CoT quality and diversity across all four sub-benchmarks. Gemini-2.5-pro attains higher CoT quality on Phys-Experiment, Phys-Problems, and Spatial-Relation, while GPT-5 slightly outperforms it on Dyn-Prediction. Diversity scores are also competitive for both models. **However, their scores still do not reach human-level performance and the failure patterns above remain clearly observable, which supports our view that  Visual Physical CoT continues to be a useful and challenging yardstick for evaluating physical visual reasoning as models improve**.
>
> $$
> \\begin{array}{l|ccc|ccc|ccc|ccc}
> \\hline
> \\textbf{Model} & & \\textbf{Phys-Experiment} & & & \\textbf{Phys-Problems} & & & \\textbf{Spatial-Relation} & & & \\textbf{Dyn-Prediction} & \\\\
>  & \\text{Quality} & \\text{Diversity} & \\text{Efficiency} & \\text{Quality} & \\text{Diversity} & \\text{Efficiency} & \\text{Quality} & \\text{Diversity} & \\text{Efficiency} & \\text{Quality} & \\text{Diversity} & \\text{Efficiency} \\\\
> \\hline
> \\text{Gemini-2.5-pro} & 62.69 & 80.00& 100.00 & 73.44 & 57.68 & 98.33 & 51.91 & 55.61 & 100.00 & 35.89 & 66.67 & 99.39 \\\\
> \\text{GPT-5} & 56.36 & 75.64 & 99.41 & 69.56 & 69.86& 88.97 & 44.56 & 44.07 & 99.17 & 48.17 & 59.58 & 96.19 \\\\
> \\hline
> \\end{array}
> $$
>
> ### **Q2. Consistency with Prior Findings on the Benefits of Multi-Image Reasoning**
>
> We appreciate the pointer and agree that our finding that “full image sequences improve performance” is conceptually consistent with prior work showing that more visual context helps multimodal reasoning. Our contribution is to show this effect specifically for visual physical processes.
>
> ### **Q3. Broader exploration of "physical content" beyond simple physics experiments**
>
> You are right that our initial version focused on relatively “simple” physics experiments and everyday Newtonian scenarios. In response to this limitation, we have recently introduced new evaluation items in three previously uncovered subdomains:
>
> 1. **Fluid Mechanics**: real-world experiments involving stochastic and continuous flow behaviors.
>
> 2. **Atomic Physics**: multi-modal reasoning samples targeting atomic structures, interactions, and transitions.
>
> 3. **Quantum Mechanics**: question-based evaluations requiring abstract reasoning over quantum states and uncertainty principles.
>
> We will clarify that our current benchmark now spans a broader set of visual physical regimes, while more complex large-scale systems (e.g., weather and earth science) remain important directions for future extensions of this benchmark.

---

### Official Review · Reviewer_8qam · 2025-11-03

**Soundness:** 1
**Presentation:** 3
**Contribution:** 2
**Rating:** 2
**Confidence:** 5

**Summary:**

This paper introduces MVPBench a benchmark to assess the ability of the multimodal large language models to reason about how physical world works (such as motion and causality). MVPBench includes different physical problem tasks, and includes assessing the reasoing steps (cots) rather than only the final answer. Each example includes multiple valid reasoning paths, and the authors propose a graph-based evaluation method that checks whether a model’s thought process follows physically consistent logic. Testing a wide range of open- and closed-source models, they find that even top systems struggle with spatial and causal understanding, and that reinforcement learning–based fine-tuning can actually make it worse.

**Strengths:**

- Evaluating the reasoning steps (and not just the final answer) is an important aspect of reasoning capabilities assessment. The authors proposed method and benchmark fills this important gap.
- Annotating and manually checking the reasoning steps is a time consuming task for more than 1k samples in the becnhmark.
- The experiment results on reinforcement learning–based fine-tuned models is interesting.

**Weaknesses:**

Although the benchmark and tasks seem interesting, I am not convinced of the actual quality of the samples, which is the most crucial aspect of a benchmark. For instance, the very first example shown in the paper (Figure 1) seems to be flawed. The correct answer is that the bus is moving downwards (so the front is at the bottom) but the step_1 in Textual CoT says "the tail is near the
bottom of the picture". Also, Step 1 in Model reasoning (GPT-4o), correctly says "The front of the
vehicle is at the bottom part of the vehicle (closer to the camera)." but is marked as incorrect.

In other examples of Figure 2, the questions do not seem to be clear, and the answers too. For instance, the Physical State example, "What results from A and B contacting C– F under gravity?" is very confusing even for humans.

Overall, although the focus area of paper is important, the execution is not satisfactory.

**Questions:**

Could you please provide more quantitive details on how the manual annotations of CoTs where performed? How many hours was spent on it, and by how many people each sample was manually cross checked?

Providing a human performance baseline for the tasks is crucial, as it will determine the actual do ability and quality of the benchmark.

**Details Of Ethics Concerns:**

This work includes a team of 31 human annotators.

---

> ### Author Response · Authors · 2025-11-21
>
> Thank you for the thoughtful comments. We address the concerns regarding sample quality and clarity in Figures 1 and 2 as follows.
>
> ### **Figure 1**
> We appreciate your careful inspection of Figure 1. You are correct that, in the original illustration, the bus is moving downward and the front should be at the bottom. The Step-1 textual CoT in the figure mistakenly stated that “the tail is near the bottom,” and as a result we also mis-marked the model’s first reasoning step.
>
> This example corresponds to item #72 in the Spatial Relations subset. The original benchmark annotation—*“the front of A is facing toward the lower, and the rear toward the upper”*—is correct in the dataset. The issue arose solely from the simplified schematic used for Figure 1, which was intended as an explanatory illustration rather than the actual benchmark item.
>
> In the revised pdf version, we have:
>
> * Redrawn the figure so that the vehicle orientation is correct,
> * Updated the human Step-1 CoT,
> * Corrected the annotation of the model’s reasoning step, and
> * Added the original GPT-4o output to the appendix (Fig. 22) for transparency.
>
> This correction is limited to the figure and **does not affect any dataset labels or experimental results**.
>
> ### **Figure 2**
>
> The example you referenced corresponds to item #66 in the Dynamic Prediction subset. The question is originally phrased as:
>
> > “When objects A and B move through the air and are placed down at the location of objects C–F, they come into contact with C–F due to gravity. What will happen?”
>
> For Figure 2, we condensed the phrasing to fit layout constraints. The shortened version unintentionally reduced clarity, making the question sound more ambiguous than in the actual benchmark.
>
> We have revised the figure text in the pdf to better preserve the original meaning:
>
> > “When A and B are placed above C–F and fall under gravity, they make contact with C–F. What will happen next?”
>
> Again, this issue concerns only the illustrative figure; the benchmark items used in evaluation are accurate.
>
> ### **Q1. Quantitative details on manual CoT annotation**
>
> The dataset was constructed between **March 28 and May 14, 2025** by **31 annotators** with backgrounds in physics and vision. MVPBench contains **1,211** samples across four subset, each with multi-step CoTs (avg. **2.93 steps** and **2.67 valid reasoning paths** per item), yielding several thousand verified reasoning steps overall.
>
> Annotators followed a unified protocol:
> (1) curate raw videos/images/problems and remove visually ambiguous items;
> (2) identify key objects/events and add necessary visual cues;
> (3) write a **primary multi-step CoT** linking visual evidence to physical events;
> (4) record **equivalent alternative paths of the answer**;
> (5) submit the full item for cross-checking.
>
> For conceptual physics problems, annotators first select questions where visual content is crucial, then add visual cues (e.g., red dots, arrows) directly on the images. Using collected reference solutions, they separately annotate textual and visual CoTs, express step-by-step reasoning (including equations) in Markdown-style math, and organize all reasoning paths into a unified JSON file (with multiple inference paths recorded separately) before submitting the final example for cross-checking.
>
> As detailed in Appendix C, all samples underwent **double annotation**: two annotators independently produced reasoning steps, followed by cross-checking and expert resolution of conflicts. Across the 7-week period, the team spent approximately **5000 hours** on writing, verifying, and cleaning CoTs—about **4.1 hours per sample** including all cross-checking stages.
>
> ### **Q2. Human performance baseline**
>
> We fully agree that human performance is crucial for assessing the actual difficulty and quality of the benchmark. Our evaluation results are reported in **Appendix B.1** and summarized in **Table 5**.
>
> Although MVPBench is framed as a visual question answering (VQA) task, its evaluation protocol requires detailed, step-by-step reasoning chains, making the **quality** and **efficiency** metrics challenging to assess for humans. Nevertheless, we conducted an additional human study to provide a more complete comparison. We randomly sampled 15 instances from each of the four major subsets and asked the same participants to produce full reasoning chains under the same constraints used in model evaluation.
>
> The results reveal clear differences across subsets. Humans achieve near-perfect performance on Spatial Relations and Dynamic Prediction, reflecting strong alignment with intuitive physical reasoning. Performance drops on Physics Experiments and Physics Problems, which demand higher conceptual understanding and more complex procedural reasoning. The detailed results are provided in **Appendix J, Table 11**.
>
> We hope this clarifies our human baseline evaluation setup.

---

### Meta-Review · Area_Chair_cNfn · 2026-01-06

**Summary:**

The paper introduces MVPBench, a benchmark designed to evaluate the visual physical reasoning capabilities of Multimodal Large Language Models (MLLMs). The key contributions include a dataset of 1,211 examples annotated with multi-path Chains of Thought (CoT) and a new graph-based metric (incorporating Graph Edit Distance) to evaluate reasoning consistency, diversity, and quality. The authors report that current state-of-the-art models struggle with these tasks and present a finding that RL-based post-training may impair spatial reasoning.

While the motivation to move beyond single-answer evaluation to reasoning path evaluation is well-received, the reviewers identified significant issues regarding the complexity and validity of the proposed metric, the quality control of the dataset, and the scope of the physical tasks.

**Reviewer Concerns:**

Addressed Concerns (or at least partially addressed):

- Errors in Figures: Reviewer pointed out a critical error in Figure 1 where the ground truth direction of a vehicle was incorrect, and ambiguities in Figure 2. The authors acknowledged this oversight, attributed it to the schematic design rather than the dataset labels, and updated the figure in the revision.

- Missing Baselines: The authors addressed the lack of human performance baselines and added results for newer models like GPT-5 (thinking) and Gemini-2.5-Pro.

- Unbalanced Data: Reviewer concern about the unbalanced nature of the dataset sub-categories was addressed by providing an analysis on a balanced subset, showing consistent trends.

Outstanding concerns:

- Metric Complexity and Validity: Reviewer expressed strong concerns regarding the proposed graph-based CoT evaluation metric. They argued that the reliance on Graph Edit Distance (GED) and multiple hand-tuned hyperparameters makes the metric excessively complex and its superiority over simpler metrics unproven. Despite the authors' rebuttal explaining the sensitivity analysis of the hyperparameters, Reviewer explicitly stated in a follow-up comment that this concern remains unsolved.

- Reliance on Model-Based Evaluation: Reviewer noted the circularity of using GPT-4o as a judge to evaluate CoT quality when GPT-4o itself is one of the models under test. While the authors tested other judges (e.g., Gemini), the fundamental concern about the reliability of LLM-based judges for fine-grained physical reasoning logic remains a hurdle for a benchmark paper claiming to be a standard.

- Dataset Quality and Trust: Reviewers identification of a factual error in the paper’s teaser figure (Figure 1) significantly undermined confidence in the dataset's quality assurance. For a benchmark paper, "Ground Truth" must be impeccable. Although the authors fixed the figure, the initial presence of such an error suggests potential issues with the verification process of the 1,211 samples.

- Scope: Reviewer noted that despite the title "Visual Physical CoT," the tasks are largely limited to simple physics experiments rather than broader complex physical systems.

**Reviewer Scores:**

- Reviewer 8qam (Score: 2): Score likely remains 2 (Reject). The reviewer found the execution unsatisfactory and the errors in the illustrative figures fundamentally undermined confidence in the benchmark's quality.

- Reviewer nCwg (Score: 6): Score likely remains 6 (Marginally Accept). This reviewer was the most positive, appreciating the insights and the new baselines provided in the rebuttal.

- Reviewer t9yx (Score: 4): Score likely improves to 5 (Marginally Below Acceptance) or remains 4. The authors addressed the specific technical points (unbalanced data, sensitivity), but given the overarching skepticism from other reviewers about the metric and data quality, this reviewer's "marginal" stance is unlikely to flip to a strong accept.

- Reviewer xjhL (Score: 4): Score likely remains 4 (Marginally Reject) or drops. The reviewer explicitly commented that the rebuttal did not resolve their core concerns regarding the metric's design and complexity.

---

### Decision · Program_Chairs · 2026-01-26

Reject